# Temporal inventory of glaciers in the Suru sub-basin, western Himalaya: Impacts of the regional climate variability

Aparna Shukla[1,2*], Siddhi Garg[1], Manish Mehta[1], Vinit Kumar[1], Uma Kant Shukla[3]

[1]Wadia Institute of Himalayan Geology, 33, GMS Road, Dehradun-248001, India
[2]Ministry of Earth Sciences, New Delhi– 110003, India
[3]Department of Geology, Banaras Hindu University, Varanasi –221005, India

[*]*Correspondence to*: Aparna Shukla (aparna.shukla22@gmail.com)

**Abstract**

Updated knowledge about the glacier extent and characteristics in the Himalaya cannot be overemphasised. Availability of precise glacier inventories in the latitudinally diverse western Himalayan region is particularly crucial. In this study we have created an inventory of the Suru sub-basin, western Himalaya for year 2017 using Landsat OLI data. Changes in glacier parameters have also been monitored from 1971 to 2017 using temporal satellite remote sensing data and limited field observations. Inventory data show that the sub-basin has 252 glaciers covering 11% of the basin, having an average slope of 25 $\pm6°$ and dominantly north orientation. The average snow line altitude (SLA) of the basin is 5011 $\pm54$ masl with smaller (47%) and cleaner (43%) glaciers occupying the bulk area. Longterm climate data (1901-2017) show an increase in the mean annual temperature ($T_{max}$ & $T_{min}$) by 0.77 ºC (0.25 & 1.3 ºC) in the sub-basin, driving the overall glacier variability in the region. Temporal analysis reveals a glacier shrinkage of ~6 $\pm0.02$%, an average retreat rate of 4.3 $\pm1.02$ ma$^{-1}$, debris increase of 62% and 22 $\pm60$ m SLA rise in past 46 years. This confirms their transitional response between the Karakoram and the Greater Himalayan Range (GHR) glaciers. Besides, glaciers in the sub-basin occupy two major ranges, i.e., GHR and Ladakh range (LR) and experience local climate variability, with the GHR glaciers exhibiting a warmer and wetter climate as compared to the LR glaciers. This variability manifestes itself in the varied response of GHR and LR glaciers. While the GHR glaciers exhibit an overall rise in SLA (GHR: 49 $\pm69$ m; LR: decrease by 18 $\pm50$ m), the LR glaciers have deglaciated more (LR: 7%; GHR: 6%) with an enhanced accumulation of debris cover (LR: 73%; GHR: 59%). Inferences from this study reveal prevalence of glacier disintegration and overall degeneration, transition of clean ice to partially debris covered glaciers, local climate variability and non-climatic (topographic and morphometric) factor induced heterogeinty in glacier response as the major processes operatives in this region. The dataset Shukla et al., (2019) is accessible at https://doi.pangaea.de/10.1594/PANGAEA.904131

**Key words:** Suru sub basin, western Himalaya, glacier inventory, climate change

**Location of the dataset:** https://doi.pangaea.de/10.1594/PANGAEA.904131

# 1 Introduction

State of the Himalayan cryosphere has a bearing on multiple aspects of hydrology, climatology, environment and sustenance of living organisms at large (Immerzeel et al., 2010; Miller et al., 2012). Being sensitive to the ongoing climate fluctutations, glaciers keep adjusting themselves and these adaptations record the changing patterns in the global climate (Bolch et al., 2012). Any alteration in the glacier parameters would ultimately affect the hydrology of the region, thereby influencing the downstream communities (Kaser et al., 2010; Pritchard, 2017). Owing to these reasons, quantifying the mass loss over different Himalayan regions in the past years, ascertaining present status of the cryosphere and how these changes are likely to affect the freshwater accessibility in the region are at the forefront of contemporary cryospheric research (Brun et al. 2017; Sakai and Fujita, 2017). This aptly triggered several regional (Kaab et al., 2012; Gardelle et al., 2013; Brun et al. 2017; Zhou et al., 2018; Maurer et al., 2019) , local (Bhushan et al., 2018; Vijay and Braun, 2018) and glacier specific studies (Dobhal et al., 2013; Bhattacharya et al., 2016; Azam et al., 2018) in the region. These studies at varying

scales contribute towards solving the jigsaw puzzle of the Himalayan cryosphere. The regional scale studies
operate on small scale for bringing out more comprehensive, holistic and synoptic spatio-temporal patterns of
glacier response, the local scale studies monitor glaciers at basin level or groups and offer more details on
heterogenous behaviour and plausible reasons thereof. However, the glacier specific studies whether based on
field or satellite or integrative information are magnified versions of the local scale studies and hold the
potential to provide valuable insights into various morphological, topographic and local-climate induced
controls on glacier evolution. Despite these efforts, data on the glacier variability and response remain
incomplete, knowledge of the governing processes still preliminary and the future viability pathways of the
Himalayan cryospheric components are uncertain.
Though the literature suggests a generalised mass loss scenario (except for the Karakoram region) over the
Himalayan glaciers, disparities in rates and pace of shrinkage remain. Maurer et al. (2019) report the average
mass wastage of -0.32 m w.e.a$^{-1}$ for the Himalayan glaciers during 1975-2016. They suggest that the glaciers in
the eastern Himalaya (-0.46 m w.e.a$^{-1}$) have experienced slightly higher mass loss as compared to the western (-
0.45 m w.e.a$^{-1}$), followed by the central (-0.38 m w.e.a$^{-1}$). However, considerable variability in the glacier
behaviour exists within the western Himalayas (Scherler et al., 2011; Kaab et al., 2012; Vijay and Braun, 2017;
Bhushan et al., 2018; Mölg et al., 2018). Studies suggest that largely the glaciers in the Karakoram Himalayas
have either remained stable or gained mass in the last few decades (Kääb et al., 2015; Cogley, 2016), while a
contrasting behaviour is observed for the GHR glaciers experiencing large scale degeneration, with more than
65% glaciers retreating during 2000-2008 (Scherler et al., 2011). However, there are two views pertaining to the
glaciers in the Trans Himalayan range, with one suggesting their intermediate response between the Karakoram
Himalaya and GHR (Chudley et al., 2017) and the other emphasizing upon their affinity either towards the GHR
or the Karakoram Himalayan glaciers (Schmidt and Nusser, 2017). Therefore, in order to add more data and
build a complete understanding of the glacier response, particularly in the western Himalaya, more local scale
studies are necessary.
Complete and precise glacier inventories form the basic prerequisites not only for comprehensive glacier
assessment but also for various hydrological and climate modelling related applications (Vaughan et al., 2013).
Information on spatial coverage of glaciers in any region is a much valued dataset and holds paramount
importance in the future assessment of glaciers. Errors in the glacier outlines may propagate and introduce
higher uncertainties in the modelled outputs (Paul et al., 2017). Besides, results from modelling studies
conducted over same region but using different sources of glacier boundaries are rendered uncomparable,
constraining the evaluation of models and thus their future development. On the other hand, quality, accuracy
and precision associated with glacier mapping and outline delineation requires dedicated efforts. Several past
studies discuss the methods for, challenges in achieving an accurate glacier inventory and resolutions for the
same (Paul et al., 2013; 2015; 2017). Thorough knowledge of glaciology and committed manual endeavour are
two vital requirements in this regard. Realisation of above facts did result in several devoted attempts to prepare
detailed glacier inventories at global scale, such as Randolph glacier inventory (RGI), Global land ice
measurements from space (GLIMS) and recently Chinese glacier inventory (CGI) and Glacier area mapping for
discharge from the Asian mountains (GAMDAM) (Raup et al., 2007; Pfeffer et al., 2014; Shiyin et al., 2014;
Nuimura et al., 2015). However, several issues related to gap areas, differences in mapping methods and skills
of the analysts involved act as limitations and need further attention.
Considering the above, present work studies the glaciers in the Suru Sub-basin (SSB), western Himalaya,
Jammu and Kashmir. Primary objectives of this study include: 1) presenting the inventory of recent glacier data
[area, length, debris cover, SLA, elevation (min & max), slope and aspect] in the SSB; 2) assessing the temporal
changes for four epochs in past 46 years; and 3) analysing the observed glacier response in relation to the
regional climate trends, local climate variability and other factors (regional hypsometry, topographic
characteristics, debris cover and geomorphic features). Several remote sensing and field based studies of
regional (Vijay and Braun, 2018) , local (Bhushan et al., 2018, Kamp et al., 2011; Pandey et al., 2011; Shukla
and Qadir, 2016, Rashid et al 2017, Murtaza and Romshoo, 2015) and glacier-specific nature (Garg et al., 2018;
2019; Shukla et al., 2018) have been conducted for monitoring the response of the glaciers to the climate
change. Glaciological studies carried out in or adjacent to the SSB suggest increased shrinkage, slowdown and
downwasting of the studied glaciers at variable rates (Kamp et al., 2011; Pandey et al., 2011; Shukla and Qadir,
2016; Bhushan et al., 2018). These studies also hint towards the possible role of topographic & morphometric
factors as well as debris cover in glacier evolution, though confined to their own specific regions. Previous
studies have also estimated the glacier statistics of SSB and reported the total number of glaciers and the
glacierized area to be 284 and 718.86 km$^2$ (Sangewar and Shukla, 2009) and 110 and 156.61 km$^2$ (SAC report,
2016), respectively. While the RGI reports varying results by two groups of analysts (number of glaciers: 514 &
covering an area of 550 & 606 km$^2$, respectively) for 2000 itself.
Previous findings suggesting progressive degeneration of glaciers, apparent variation and discrepancies in
inventory estimates and also the fact that the currently available glacier details for the sub-basin are nearly 20
years old,  mandate the recent and accurate assessment of the glaciers in the SSB and drive the present study.

## 2 Study area

The present study focuses on the glaciers of the SSB situated in the state of Jammu and Kashmir, western
Himalaya (Fig. 1). The geographic extent of the study area lies within latitude and longitude of 33° 50' to 34°
40' N to 75° 40'to 76° 30' E.
Geographically, the sub-basin covers part of two major ranges, i.e., GHR and LR and shows the presence of the
highest peaks of Nun (7135 masl) and Kun (7077 masl) in the GHR (Vittoz, 1954). The glaciers in these ranges
have distinct morphology, with the larger ones located in the GHR and comparatively smaller towards the LR
(Fig. 1).

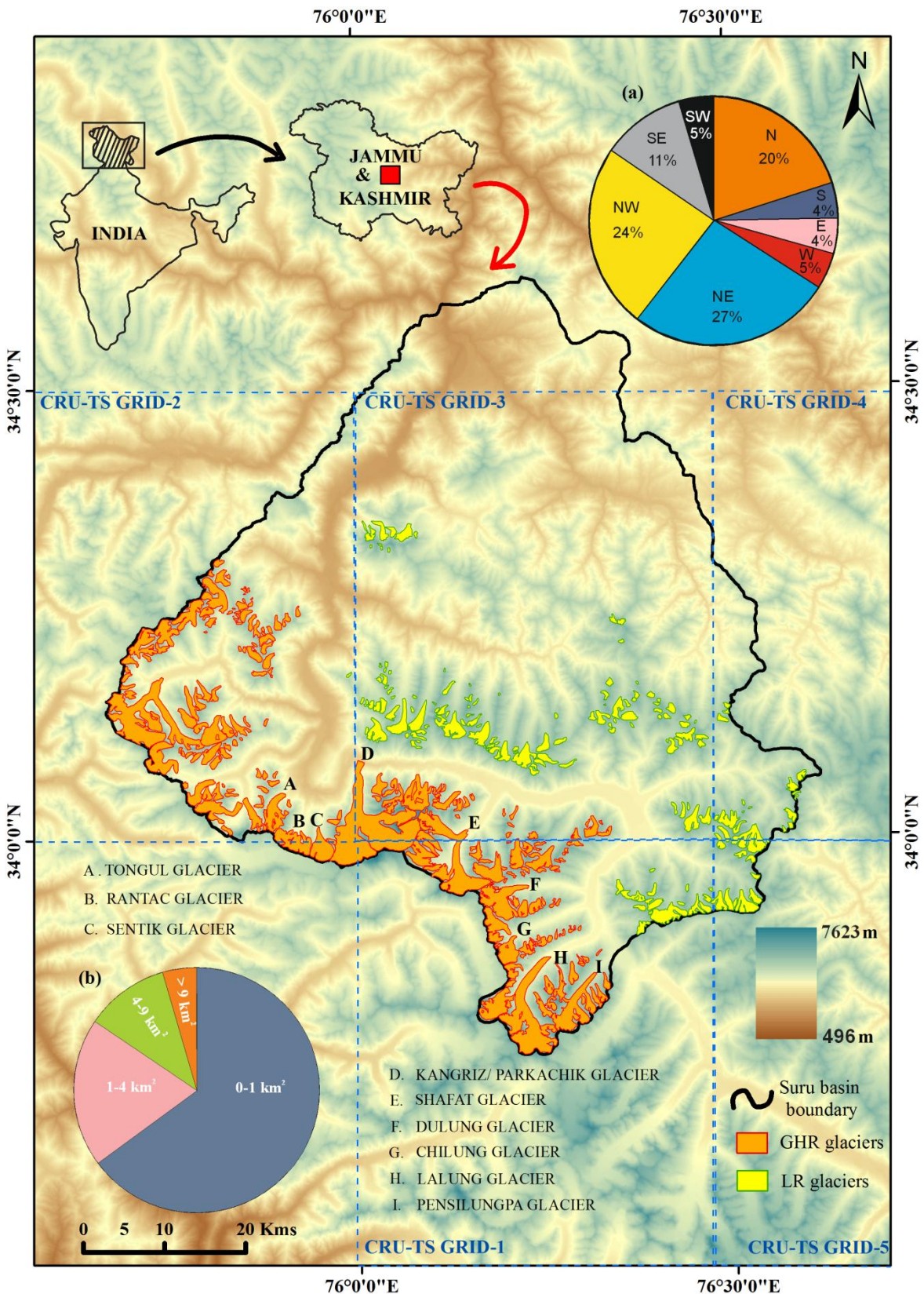

Figure 1: Location map of the study area. The glaciers in the Suru Sub-basin (black outline) are studied for their response towards the climatic conditions during the period 1971-2017. Blue rectangles with dashed outlines (GRID-1, 2, 3, 4 and 5) are the Climate Research Unit (CRU)-Time Series (TS) 4.02 grids of dimension 0.5° x 0.5°.(a) Pie-chart inset showing orientation-wise percentage distribution of glaciers in the sub-basin. North (N), north-east (NE), north-west (NW), south (S), south-east (SE), south-west (SW), east (E) and west (W)

represents the direction of the glaciers. (b) Pie chart inset showing size-distribution of glaciers in the SSB. The
glacier boundaries [GHR (orange) and LR (yellow)] are overlain on the Advanced Land Observing Satellite
(ALOS) Digital Surface Model (DSM).

The meltwater from these glaciers feeds the Suru River (tributary of Indus River), which emerges from the
Pensilungpa glacier (Fig. 2a) at an altitude of ~4675 m asl. The river further flows north for a distance of ~24
kms and takes a westward turn from Rangdum (~4200 m asl). While flowing through this path, the Suru River is
fed by some of the major glaciers of the GHR namely Lalung, Dulung (Fig. 1), Chilung (Fig. 2b), Shafat (Fig.
2c; d), Kangriz/ Parkachik (Fig. 2e), Sentik, Rantac (Fig.2f), Tongul (Fig. 2g) and Glacier no.47 (Fig. 2h).
Amongst these major glaciers, Kangriz forms the largest glacier in the SSB, covering an area of ~53 $km^2$ and
descends down from the peaks of Nun and Kun (Garg et al., 2018). The Suru River continues to flow for a
distance of nearly 54 kms and after crossing a mountain spur and the townships of Tongul, Panikhar and
Sankoo, the river further flows north until it finally merges with River Indus at Nurla (~3028 m asl).

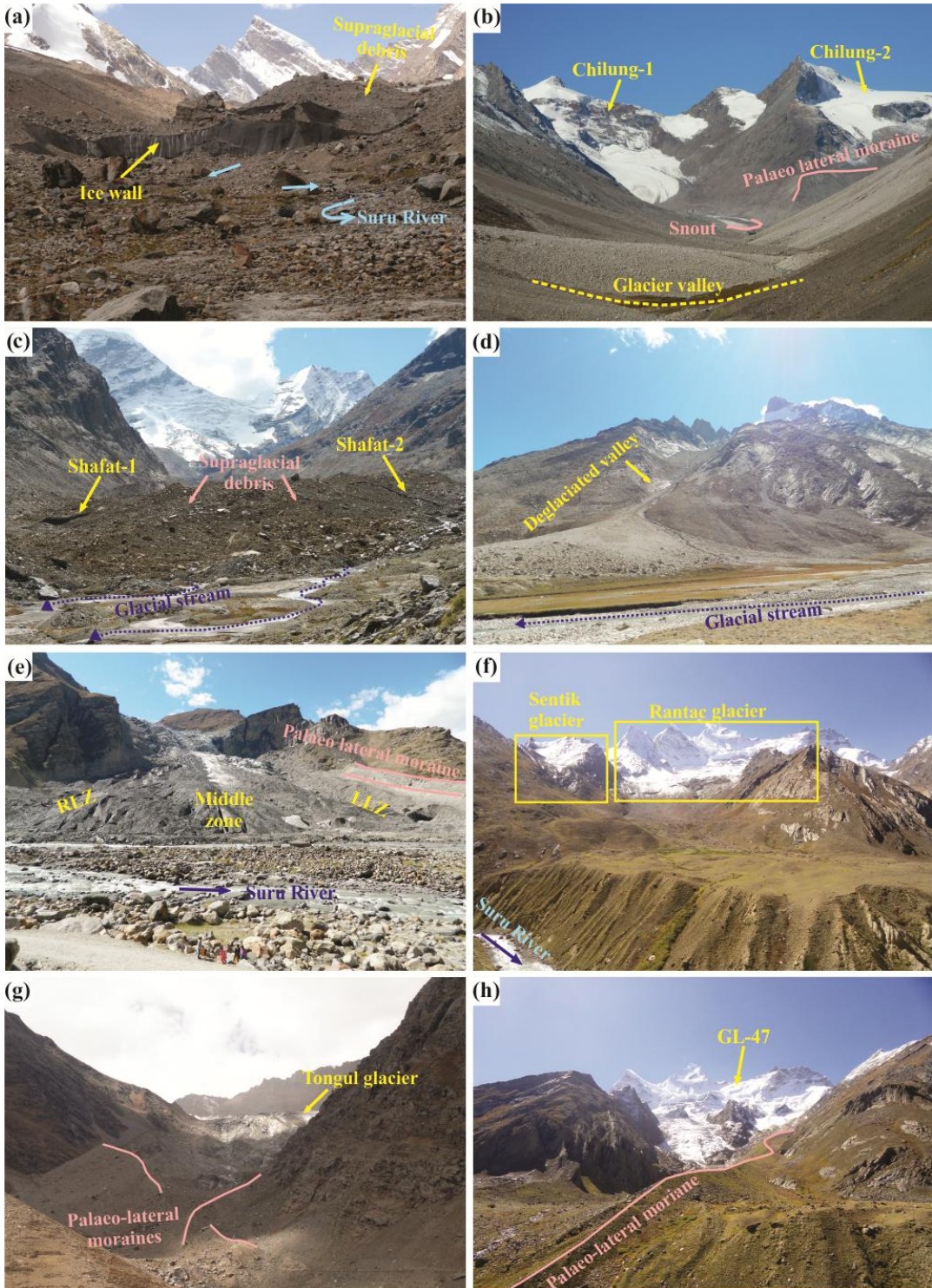

Figure 2: Field photographs of some of the investigated glaciers in the study area captured during the field visits in September, 2016 and 2017. (a), (b), (c), (e), (f), (g), (h) Snouts of Pensilungpa, Chilung, Shafat, Kangriz, Sentik & Rantac, Tongul glaciers and Glacier no.47, respectively. (d) Deglaciated valley near the Shafat glacier.

The westerlies are an important source of moisture in this region (Dimri, 2013) with wide range of fluctuations in snowfall during winters. In the Padum valley, annual mean precipitation (Snowfall) and temperature amounts to nearly 2050 to 6840 mm and 4.3 °C, respectively (Raina and Kaul, 2011; http://en.climate-data.org). The longterm average annual temperature and precipitation have varied from 5.5 °C/ 588.77 mm (Kargil) to -2.04

°C/ 278.65 mm in Leh during the period 1901-2002 (IMD, 2015). However, in order to understand the long term
variability of climatic conditions in the SSB, we have utilized the Climate Research Unit (CRU)-Time Series
(TS) 4.02 data during the period 1901-2017 (Fig. 3; Harris and Jones, 2018). Derived from this data, the annual
mean temperature and precipitation of the SSB for the period 1901-2017 has been 0.99 ±*0.45* °C and 393 ±*76*
mm, respectively. (Standard deviations associated with the mean temperature and precipitation have been
italicized throughout the text).

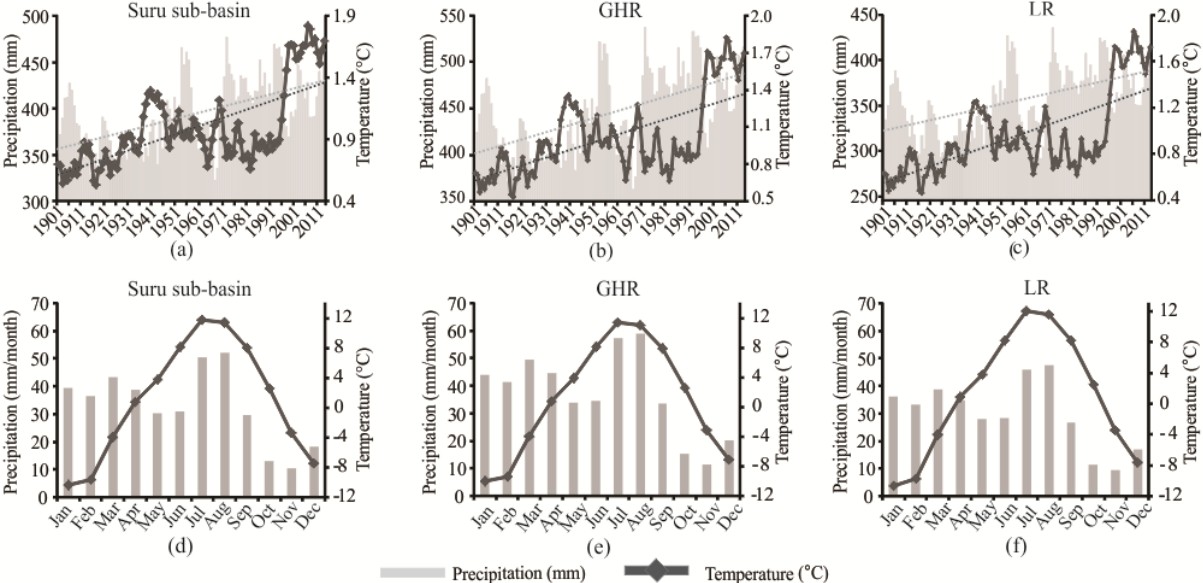

Figure 3: Annual and seasonal variability in the climate data for the period 1901-2017. (a), (b) and (c) 5 year
moving average of the mean annual precipitation (mm) and temperature (°C) recorded for 5 grids covering the
glaciers in the entire SSB, GHR and LR (sub-regions), respectively during the period 1901-2017. The light and
dark grey colored dashed lines depict the respective trend lines for precipiatation and temperature conditions
during the period 1901-2017. (d), (e) and (f) Monthly mean precipitation and temperature data for the entire
SSB, GHR and LR (sub-regions), respectively for the time period 1901-2017.

## 3 Datasets and Methods

### 3.1 Datasets used

The study uses multi-sensor and multi-temporal satellite remote sensing data for extracting the glacier
parameters for four time periods, i.e., 1971/1977, 1994, 2000 and 2017, details of which are mentioned in Table
1. It involves 6 Landsat level 1 terrain corrected (L1T), 3 strips of declassified Corona KH-4B and 1 Sentinel
multispectral scenes, downloaded from USGS Earth Explorer (https://earthexplorer.usgs.gov/). Besides, a global
digital surface model (DSM) dataset utilizing the data acquired by the Panchromatic remote-sensing Instrument
for Stereo Mapping (PRISM) onboard the Advanced Land Observing Satellite (ALOS) have also been
incorporated (https://www.eorc.jaxa.jp/ALOS/en/aw3d30/). ALOS World 3D comprises of a fine resolution
DSM (approx 5m vertical accuracy). It is primarily used for delineating the basin boundary, extraction of SLA,
elevation range, regional hypsometry and slope.

Table 1: Detailed specifications of the satellite data utilised in the present study. GB= glacier boundaries,
DC=debris cover

| S. no | Satellite sensors(Date of acquisition) | Remarks on quality | Scene Id | RMSE error | Registration accuracy (m) | | Purpose |
|---|---|---|---|---|---|---|---|
| 1. | Corona KH-4B (28 Sep 1971) | Cloud free | DS1115-2282DA056/ DS1115-2282DA055/ DS1115-2282DA054 | 0.1 | 0.3 | | Delineation of GB |
| 2. | LandsatMSS (19 Aug 1977/ 1 Aug 1977) | Cloud free/ peak ablation (17 Aug) | LM02_L1TP_159036 _19770819_20180422 _01_T2/ LM02_L1TP_159036 _19770801_20180422 _01_T2 | 0.12 | 10 | | Delineation of GB, SLA&DC |
| 3. | LandsatTM (27 Aug 1994) | Partially cloud covered/ peak ablation | LT05_L1TP_148036_ 19940827_20170113_ 01_T1/ LT05_L1GS_148037_ 19940827_20170113_ 01_T2 | 0.22 | 6 | | Delineation of GB, SLA&DC |
| 4. | LandsatTM (26 July 1994) | Seasonal snow cover | LT05_L1TP_148036_ 19940726_20170113_ 01_T1 | 0.2 | 6 | | Delineation of GB |
| 5. | LandsatETM$^+$ (4 Sep 2000) | Cloud free/ peak ablation | LE71480362000248S GS00 | Base image | | | Delineation of GB, SLA& DC |
| 6. | LandsatOLI (25July 2017) | Partially cloud covered/ peak ablation | LC08_L1TP_148036_ 20170810_01_T1 | 0.15 | | 4.5 | Delineation of GB & DC, estimation of SLA |
| 7. | Sentinel MSI (20 Sep 2017) | Cloud free | S2A_MSIL1C_20170 920T053641_N0205_ R005_T43SET_20170 920T053854 | 0.12 | | 1.2 | Delineation of GB & DC |
| 8. | LISS IV (27Aug2017 ) | Cloud free | 183599611 | 0.2 | | 1.16 | Accuracy assessment |

The aforementioned satellite images were acquired keeping into consideration certain necessary pre-requisites,
such as, peak ablation months (July/ August/ September), regional coverage, minimal snow and cloud cover for
the accurate identification and demarcation of the glaciers. Only three Corona KH-4B strips were available for
period 1971, which covered the SSB partially, i.e., 40% of the GHR and 57% of the LR glaciers. Therefore, rest
of the glaciers were delineated using the Landsat MSS image of the year 1977 (Table 1). Similarly, some of the
glaciers could not be mapped using the Landsat TM image of 27 Aug 1994 as the image was partially covered
with clouds. Therefore, 26 July 1994 image of the same sensor was used in order to delineate the boundaries of
the cloud covered glaciers.
Besides, long term climate data have been obtained from CRU-TS 4.02, which is a high resolution gridded
climate dataset obtained from the monthly meteorological observations collected at different weather stations of
the World. In order to generate this long term data, station anomalies from 1961-1990 are interpolated into 0.5°
latitude and longitude grid cells (Harris and Jones, 2018). This dataset includes six independent climate
variables (mean temperature, diurnal temperature range, precipitation, wet-day frequency, vapour pressure and
cloud cover). However, in this study monthly mean, minimum and maximum temperature and precipitation data
are taken into consideration.

## 3.2 Methodology adopted

The following section mentions the methods adopted for data extraction, analysis and uncertainty estimation.

### 3.2.1 Glacier mapping and estimation of glacier parameters

Initially, the satellite images were co-registered by projective transformationat at sub-pixel accuracy with the
Root Mean Square Error (RMSE) of less than 1m (Table 1), taking the Landsat ETM$^+$ image and ALOS DSM
as reference. However, the Corona image was co-registered following a two step approach: (1) projective
transformation was performed using nearly 160-250 GCPs (2) spline adjustment of the image strips (Bhambri et
al., 2012). The glaciers were mapped using a hybrid approach, i.e., normalized difference snow index (NDSI)
for delineating snow-ice boundaries and manual digitization of the debris cover. Considering that not many
changes would have occurred in the accumulation region, major modifications have been done in the boundaries
below the equilibrium line altitude (ELA) (Paul et al., 2017). The glacierets/ tributary glaciers contributing to
the main trunk are considered as single glacier entity. NDSI was applied on a reference image of Landsat ETM$^+$
using an area threshold range of 0.55-0.6. A median filter of kernel size 3*3 was used to remove the noise and
very small pixels. In this manner, glaciers covering a minimum area of 0.01 km$^2$ have been mapped. However,
some pixels of frozen water, shadowed regions were manually corrected. Thereafter, the debris covered part of
the glaciers was mapped manually by taking help from slope and thermal characteristics of the glaciers. Besides,
high resolution imageries from the Google Earth$^{TM}$ were also referred for the accurate demarcation of the
glaciers. Identification of the glacier terminus was done based on the presence of certain characteristic features
at the snout such as ice wall, proglacial lakes and emergence of streams. Length of the glacier was measured
along the central flow line (CFL) drawn from the bergschrund to the snout. Fluctuations in the snout position
(i.e., retreat) of an individual glacier was estimated using the parallel line method, in which parallel strips of 50
m spacing are taken on both sides of the CFL. Thereafter, the average values of these strips intersecting the
glacier boundaries were used to determine the frontal retreat of the glaciers (Shukla and Qadir, 2016; Garg et al.,
2017a;b). Mean SLA estimated at the end of the ablation season can be effectively used as a reliable proxy for
mass balance estimation for a hydrological year (Guo et al., 2014). The maximum spectral contrast between
snow and ice in the SWIR and NIR bands helps in delineation of the snow line separating the two facies. The
same principle was used in this study to yield the snow line. Further, a 15 m sized buffer was created on both
sides of the snow line to obtain the mean SLA. Other factors such as elevation (max & min), regional
hypsometry and slope were extracted utilising the ALOS DSM.

### 3.2.2 Analysis of climate variables


To ascertain the long term climate trends in the sub-basin, mean annual temperature (min & max) and
precipitation are derived by averaging the mean monthly data of the respective years. Besides, seasonal trends
are also analysed for winter (November-March) and summer (April-October) months. Moreover, the climate
variables are assessed separately for the ~46 year period (1971-2017), which is the study period of present
research.
Further, the climate dataset was statistically analysed for five grids using Mann-Kendall test to obtain the
magnitude and significance of the trends (Supplementary table S2). The magnitude of trends in time series data
was determined using Sen's slope estimator (Sen, 1968). Quantitatively, the temperature and precipitation trends
have been assessed here in absolute terms (determined from Sen's slope). The change in climate parameters
(temperature and precipitation) was determined using following formula:
$$\text{Change} = (\beta * L) / M \qquad (1)$$
where $\beta$ is Sen's slope estimator, $L$ is length of period and $M$ is the long term mean.
These tests were performed at confidence level, S= 0.1(90%), 0.05( 95%) and 0.01(99%), which differed for
both the variables (Supplementary table S2). Spatial interpolation of climate data was achieved using the Inverse
Distance Weighted (IDW) algorithm. For this purpose, a total number of 15 CRU TS grids (in vicinity of our
study area) were taken so as to have an ample number of data points in order to achieve the accurate results.
Further, in order to check data consistency, we have taken instrument data from nearest stations of Kargil and
Leh (due to the unavailability of meteorological stations in the Suru sub-basin) and compared with the CRU-TS
derived data for the entire Suru sub-basin during 1901-2002 period (Fig. 4).

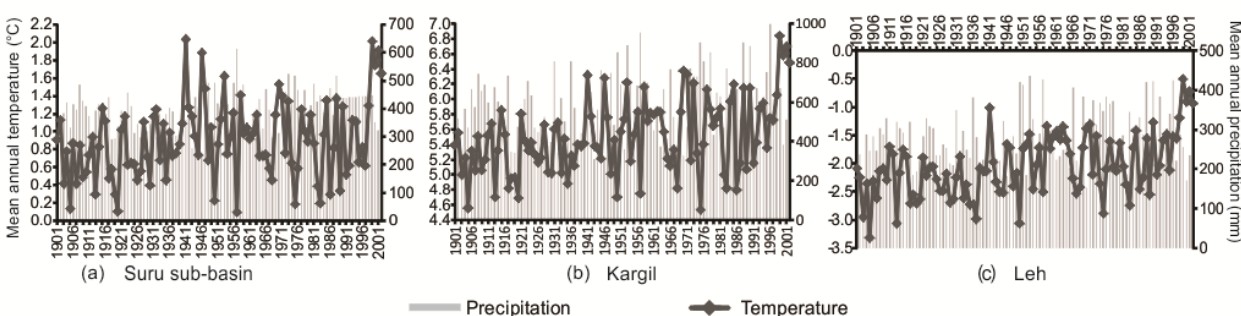

Figure 4: Mean annual temperature and precipitation patterns of CRU-TS derived gridded data in (a) Suru sub-
basin and IMD recorded station at (b) Kargil and (c) Leh.
The mean annual temperature pattern of Suru sub-basin shows a near negative trend till 1937, with an increase
thereafter. Similar trends have been observed for Kargil and Leh, despite their distant location from the Suru
sub-basin (areal distance of Kargil and Leh is ~63 and 126 km, respectively from the centre of Suru sub-basin).
However, it is noteworthy to mention that all the locations had attained maximum mean annual temperature in
1999 (Suru: 2.02°C; Kargil: 6.84°C; Leh: -0.5°C). We observe an almost similar trend in all the cases (Fig. 4),
with an accelerated warming post 1995/96. However, the magnitude varies, with longterm mean annual
temperature of 0.9, 5.5 and -2.04°C observed in Suru sub-basin, Kargil and Leh, respectively (Fig. 4). The
possible reason for this difference in their magnitudes could possibly be attributed to their distinct geographical
locations and difference in their nature, with former being point, while latter being the interpolated gridded data.
Also, we have used the station data, obtained from nearest available IMD sites, i.e., Kargil and Leh and
compared with their respective CRU-TS data (mean annual temperature and precipitation).

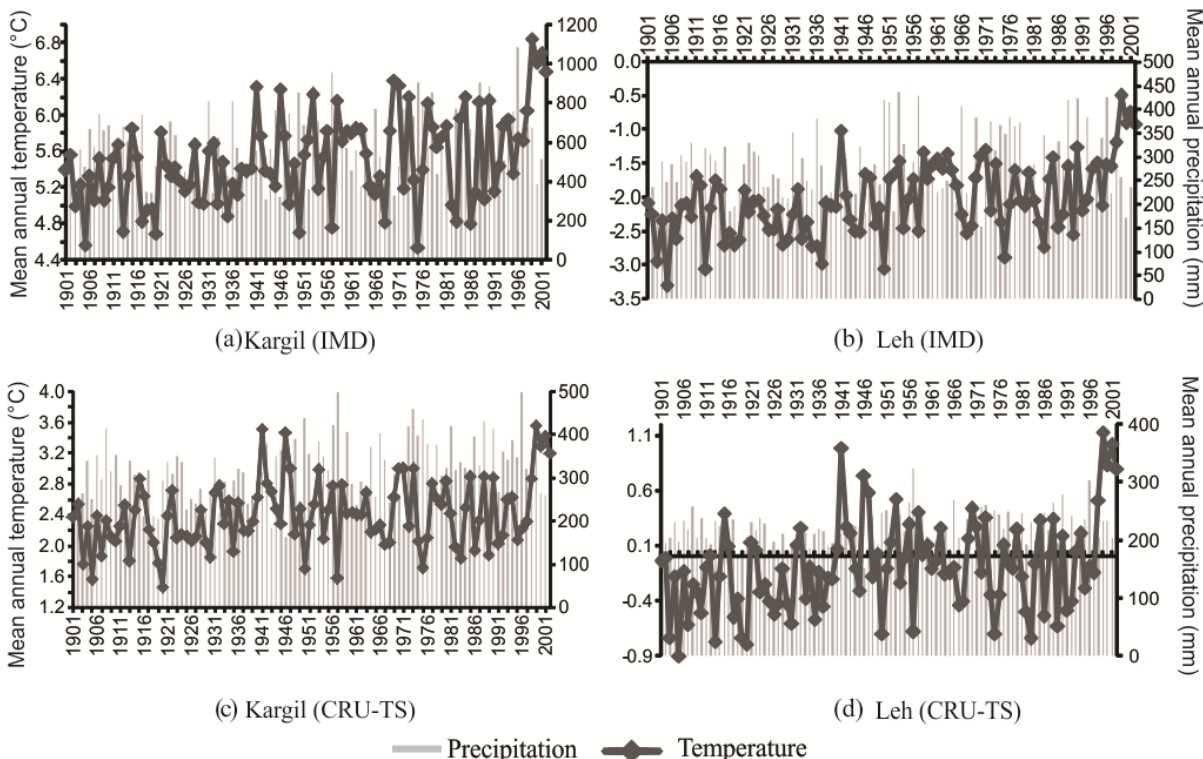

Figure 5: Analysis of meteorological (mean annual temperature and precipitation) datasets derived from Indian
Meteorological Department (IMD) stations at (a) Kargil & (b) Leh and the respective [(c) Kargil and (d) Leh]
gridded data obtained from climate research unit (CRU)-time series (TS).
Though varying in magnitude, the climate data obtained from IMD as well as CRU-TS suggest almost similar
trends of temperature and precipitation during the period 1901-2002 for both Kargil and Leh (Fig. 5). The
annual mean temperature/ precipitation amounted to 5.5°C/589 mm (IMD) and 2.4°C/315 mm (CRU-TS) in
Kargil, while -2.04/279 mm (IMD) and -0.09/ 216 mm (CRU-TS) in Leh during the period 1901-2002 (Fig.
5).We observed that climatic variables show lower magnitude in case of CRU-TS as compared to the station
data from IMD (except CRU-TS derived temperature data recorded for Leh). The possible reason for this
difference between CRU-TS and station data can primarily be attributed to the difference in their nature, with
former being point, while latter being a gridded data (0.5° latitude and longitude grid cells). This analysis aptly
brings out the bias in the CRU TS gridded data. Majorly the comparison shows that though the gridded data
correctly bring out the temporal trends in meteorological data, but differ with station data in magnitude (being
on lower side than the station estimates). This helps us better appreciate the climate variations in the Suru sub-
basin as well, since we learn that the reported temperature and precipitation changes are probably on the lower
side of the actual variations.
**3.2.3 Uncertainty assessment**
This study involves extraction of various glacial parameters utilizing satellite data with variable characteristics,
hence, susceptible to uncertainties, which may arise from various sources. These sources may be locational
(LE), interpretational (IE), classification (CE) or processing (PE) errors (Racoviteanu et al., 2009; Shukla and
Qadir, 2016). In our study, the LE and PE may have resulted on account of miss-registration of the satellite
images and inaccurate mapping, respectively. While IE and CE would have introduced due to the miss-
interpretation of glacier features during mapping. The former can be rectified by co-registration of the images
and estimation of sub-pixel co-registration RMSE (Table 1) and using standard statistical measures. However,
the latter can be visually identified and corrected but difficult for exact quantification owing to lack of reliable
reference data (field data) in most cases. As a standard procedure for uncertainty estimation, glacier outlines are
compared directly with the ground truth data as acquired using a Differential Global Positioning System (DGPS)
(Racoviteanuet al., 2008a). In this study, DGPS survey was conducted on the Pensilungpa and Kangriz glaciers
at an error of less than 1cm. Therefore, by comparing the snout position of Pensilungpa (2017) and Kangriz
(2018) glaciers derived from DGPS and OLI image, an accuracy of ±23 and ±1.4 m, respectively was obtained.
Also, the frontal retreat estimated for the Kangriz glacier using DGPS and OLI image is found to be 38.63 ±47.8
and 39.98 ±56.6 m, respectively during the period 2017-18. In this study, high resolution Linear imaging self-
scanning system (LISS)-IV imagery (spatial resolution of 5.8 m) is also used for validating the glacier mapping
results for the year 2017 (Table 1). Glaciers of varying dimensions and distribution of debris cover were
selected for this purpose. The area and length mapping accuracy for these selected glacier boundaries (G-1, G-2,
G-3, G-13, G-41, G-209, G-215, G-216, G-220, G-233) was found to be 3% and 0.5%, respectively.
The multi-temporal datasets were assessed for glacier length and area change uncertainty as per the methods
given by Hall et al. (2003) and Granshaw and Fountain (2006). Following formulations (Hall et al., 2003) were
used for estimation of the said parameters:
$$\text{Terminus uncertainty } (U_T) = \sqrt{a^2 + b^2} + \sigma \quad (2)$$
where, 'a' and 'b' are the pixel resolution of image 1 and 2, respectively and 'σ' is the registration error. The
terminus and areal uncertainty estimated are given in Table 2.
$$\text{Area change uncertainty } (U_A) = 2 * U_T * x \quad (3)$$
where, 'x' is the spatial resolution of the sensor.
Table 2. Terminus and Area change uncertainty associated with satellite dataset as defined by Hall et al. (2003).
$U_T$ = terminus uncertainty, $U_A$= area change uncertainty, x= spatial resolution, σ = registration accuracy.

| Serial no. | Satellite sensor | Terminus uncertainty $U_T$ $=\sqrt{a^2 + b^2} + \sigma$ | Area change uncertainty $U_A = 2\,U_T * x$ |
|---|---|---|---|

| 1. | Corona KH-4B | 3.12 m | 0.00007 km$^2$ |
|---|---|---|---|
| 2. | Landsat MSS | 123.13 m | 0.03 km$^2$ |
| 3. | Landsat TM | 41.42 m | 0.003 km$^2$ |
| 4. | Landsat ETM$^+$ | 48.42 m | 0.003 km$^2$ |
| 5. | Landsat OLI | 46.92 m | 0.003 km$^2$ |

Area mapping uncertainty was estimated using the buffer method, in which, a buffer size equal the registration
error of the satellite image was taken into consideration (Bolch et al., 2012; Garg et al., 2017a,b). Error
estimated using this method is found to be 0.48, 27.2, 9.6 and 3.41 km$^2$ for the 1971 (Corona), 1977 (MSS),
1994 (TM) and 2017 (OLI) image, respectively. Since the debris extents were delineated within the respective
glacier boundaries, the proportionate errors are likely to have propagated in debris cover estimations which were
estimated accordingly (Garg et al., 2017b).
Uncertainty in SLA estimation needs to be reported in the X, Y and Z directions. In this context, error in X and
Y directions should be equal to the distance taken for creating the buffer on either side of the snow line
demarcating the snow and ice facies. Since, the buffer size taken in this study was 15 m, therefore, error in X
and Y direction was considered as ±15 m. However, uncertainty in Z direction would be similar to the ALOS
DSM, i.e., ±5 m.

**4 Results**
The present study involved creation of glacier inventory for the year 2017 and estimation of glacier (area,
length, debris cover and SLA) parameters for four different time periods. For detailed insight, the variability of
the glacier parameters have also been evaluated on decadal scale, in which the total time period has been sub-
divided into three time frames, i.e., 1971-1994 (23 years), 1994-2000 (6 years) and 2000-2017 (17 years).

**4.1 Basin statistics**
The SSB covers an area of ~4429 km$^2$. In 1971, the sub-basin had around 240 glaciers, with 126 glaciers located
in the GHR and 114 in the LR, which remained the same till 2000. However, a major disintegration of glaciers
took place during the period 2000-2017, which resulted into the breakdown of about 12 glaciers into smaller
glacierets. The recent (2017) distribution of the glaciers in the GHR and LR is 130 and 122, respectively
(Supplementary table S1). The overall glacierized area is ~11%, with the size and length of the glaciers varying
from 0.01 to 53.1 km$^2$ and 0.15 to 16.34 km, respectively.
Within the sub-basin, the size range of glaciers in the GHR and LR vary from 0.01 (G-115) to 53.1 km$^2$ (G-50)
and 0.03 (G-155/165) to 6.73 km$^2$ (G-209), respectively. Considering this, glaciers have been categorized into
small (0-7 km$^2$/ 0-2 km), medium (7-15 km$^2$/ 2-7 km) and large (>15 km$^2$/ >7 km). Based on size distribution,
small (comprising all the LR and some GHR glaciers), medium and large glaciers occupy 47%, 15% and 38% of
the glacierized sub-basin. Depending upon the percentage area occupied by the supraglacial debris out of the
total glacier area, the glaciers have been categorized into clean (CG: 0-25%), partially debris-covered (PDG: 25-
50%) and heavily debris-covered (HDG: >50%). Categorization of the glaciers based on this criteria shows their
proportion in the glacierized basin as: CG (43%), PDG (40%) and HDG (17%). Majority of the glaciers in the

sub-basin are north facing (N/ NW/NE: 71%), followed by south (S/ SW/ SE: 20%), with very few oriented in other (E/ W: 9%) directions (Fig. 1a). The mean elevation of the glaciers in the SSB is 5134.8 ±*225* masl, with an average elevation of 5020 ±*146* and 5260 ±*117* masl in the GHR and LR, respectively. Mean slope of the glaciers is 24.8 ±*5.8*° and varies from 24 ±*6*° to 25 ±*6*° in the GHR and LR, respectively. While, percentage distribution of glaciers shows that nearly 80% of the LR glaciers have steeper slope (20-40°) as compared to the GHR glaciers (57%).

## 4.2 Area changes

The glaciated area reduced from 513 ±14 km$^2$ (1971) to 481 ±3.4 km$^2$ (2017), exhibiting an overall deglaciation of 32 ±9 km$^2$ (6 ±0.02%) during the period 1971-2017. Percentage area loss of the individual glaciers ranges between 0.8 (G-50; Parkachik glacier) - 45 (G-81) %, with majority of the glaciers undergoing an area loss in the range 6-12% during the period 1971-2017 (Fig.6a).

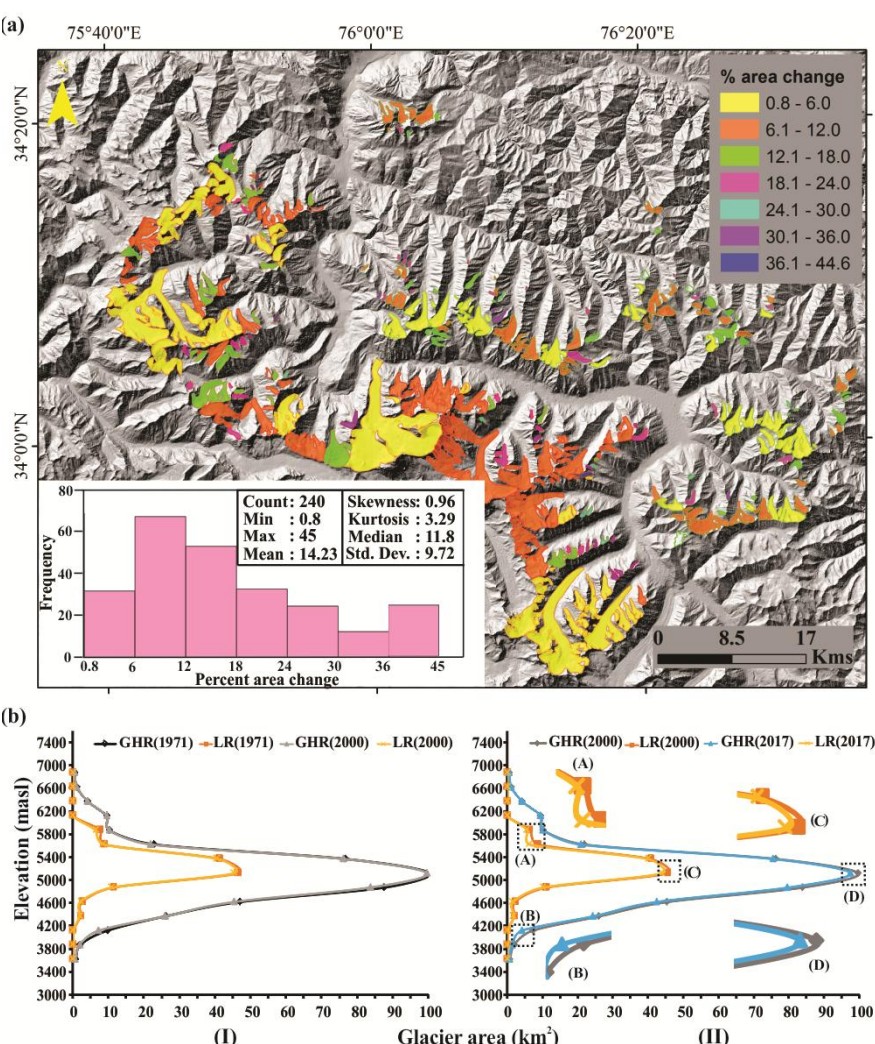

Figure 6: (a) Percent area loss of the glaciers in the SSB during the period 1971-2017. Frequency distribution histogram depicting that majority of the glaciers have undergone an area loss in the range 6-12%. (b) Hypsometric distribution of glacier area in the GHR and LR regions during the period (I) 1971-2000 and (II) 2000-2017. (A), (B), (C) and (D) insets in (II) shows the significant change in area at different elevation range of the GHR and LR glaciers.

Results show that the highest pace of deglaciation is observed during 1994-2000 (0.95 ±0.005 km$^2$a$^{-1}$) and 2000-
2017 (0.86 ±0.0002 km$^2$a$^{-1}$) followed by 1971-1994 (0.5 ±0.001 km$^2$a$^{-1}$) (Supplementary figure S1a).Within the
SSB, glaciers in the LR exhibit higher deglaciation (7 ±7.2%) as compared to GHR (6 ±2%) during the period
1971-2017. Apart from deglaciation, G-50 also showed increment in glacier area during the period 1994-2000,
however, insignificantly.

**4.3 Length changes**
Fluctuations in the glacier snout have been estimated during the period 1971-2017 and it is observed that nearly
all the glaciers have retreated during the said period, however the retreat rates vary considerably. The overall
average retreat rate of the glaciers is observed to be 4.3 ±1.02 ma$^{-1}$ during the period 1971-2017. Percentage
length change of the glaciers ranges between 0.9 to 47%, with majority of the glaciers retreating in the range  6-
14% during the period 1971-2017 (Fig.7).

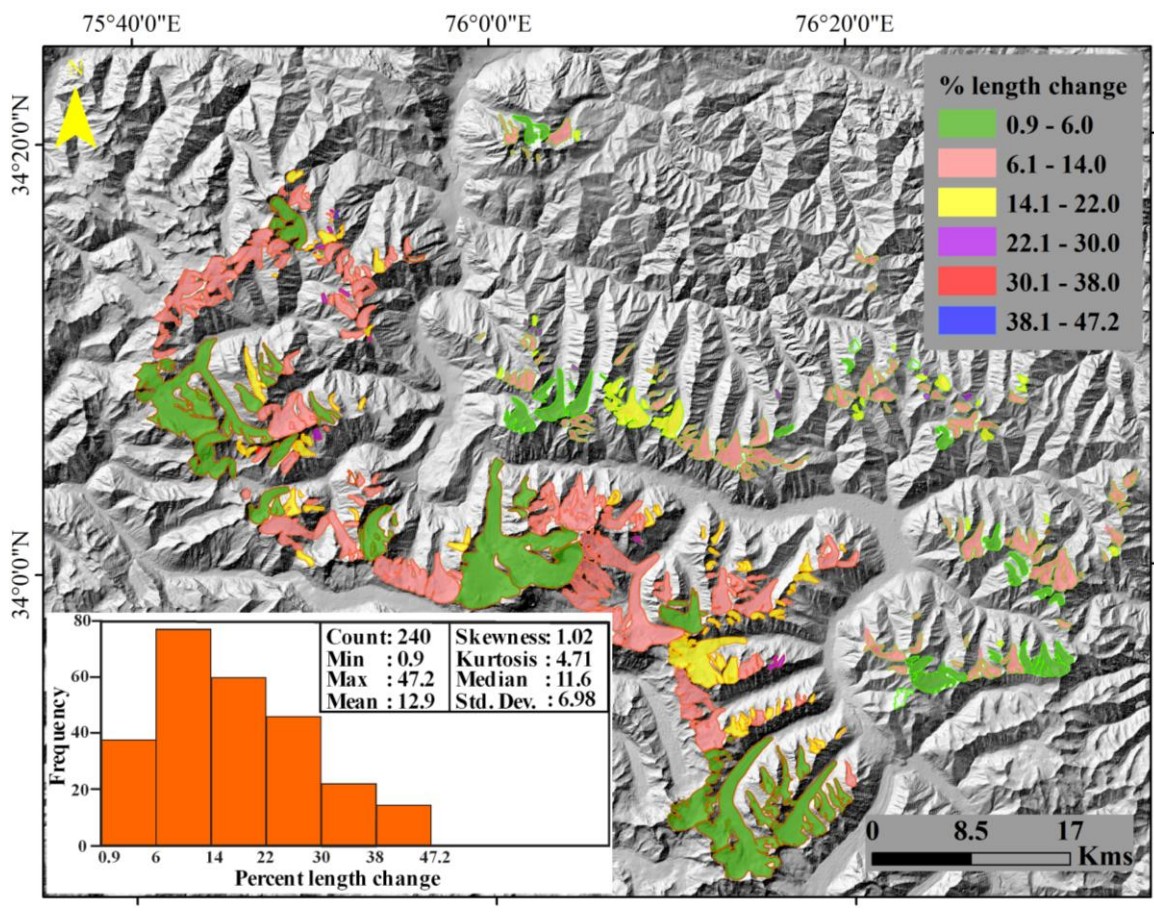


Figure 7: Percent length change of the glaciers in the SSB during the period 1971-2017. Frequency distribution
histogram showing that majority of the glaciers have undergone length change of in the range 6-14%.

Decadal observations reveal the highest rate of retreat during 1994-2000 (7.37 ±8.6 ma$^{-1}$) followed by 2000-
2017 (4.66 ±1.04 ma$^{-1}$) and lowest during 1971-1994 (3.22 ±2.3 ma$^{-1}$) (Supplementary figure S 1b). Also, the
average retreat rate in the GHR and LR glaciers was observed to be 5.4 ±1.04 ma$^{-1}$ and 3.3 ±1.04 ma$^{-1}$,
respectively, during the period 1971-2017. The retreat rate of individual glaciers varied from 0.72 ±1.02 ma$^{-1}$
(G-114) to 28.92 ±1.02 ma$^{-1}$ (G-7, i.e., Dulung glacier) during the period 1971-2017. Besides, the Kangriz
glacier (G-50) also showed advancement during the period 1994-2000 by 5.23 ±8.6 ma$^{-1}$.

**4.4 Debris-cover changes**

Results show an overall increase in debris-cover extent by 62% (~37 ±0.002 km$^2$) in the SSB glaciers during the
period 1971-2017. Decadal variations exhibit the maximum increase in the debris-cover by approximately 19
±0.00004 km$^2$ (24%) during 2000-2017 followed by an increase of 13 ±0.0001 km$^2$ (20%) and 5 ±0.0001 km$^2$
(9%) during 1994-2000 and 1971-1994, respectively (Supplementary figure S1c). However, GHR and LR
glaciers show an overall increase of debris cover extent by 59% and 73%, respectively during the entire study
period, i.e., 1971-2017.

**4.5 SLA variations**

The mean SLA shows an average increase of 22 ±*60* mduring the period 1977-2017. On the decadal scale, SLA
variations showed the highest increase (161 ±*59* m) during 1994-2000 with a considerably lower increase (8 ±*59*
m) during 1977-1994 and decrease (150 ±*60* m) during 2000-2017. Amongst the four time periods (1977, 1994,
2000 & 2017) used for mean SLA estimation, the highest SLA is noted during 2000 (5158 ±*65* masl) and
minimum during 1977 (4988 ±*65* masl) (Supplementary figure S1d).
During the period 1977-2017, the average SLA of the LR glaciers is observed to be relatively higher (5155 ±*7*
masl) as compared to the GHR glaciers (4962 ±*9* masl). In contrast, an overall rise in mean SLA was noted in
GHR (49 ±*69* m), while a decrease in LR glaciers (18 ±*45* m) during the time frame of 1977-2017.

**5 Discussion**

The present study reports detailed temporal inventory data of the glaciers in the SSB considering multiple
glacier parameters, evaluates the ensuing changes for ascertaining the status of glaciers and relates them to
climate variability and other inherent terrain characteristics. The results suggest an overall degeneration of the
glaciers with pronounced spatial and temporal heterogeneity in response.

**5.1 Glacier variability in Suru sub-basin: A comparative evaluation**

Basin statistics reveal that in the year 2000, the SSB comprised of 240 glaciers covering an area of
approximately 496 km$^2$. However, these figures differ considerably from the previously reported studies in this
particular sub-basin, with the total number of glaciers and the glacierized area varying from 284/ 718.86 km$^2$
(Sangewar and Shukla, 2009) to 110/ 156.61 km$^2$ (SAC report, 2016), respectively. In contrast, the glacierized
area is found to be less, however comparable with the RGI boundaries (550.88 km$^2$). Besides, debris cover
distribution of the glaciers during 2000 is observed to be ~16% in the present study,which is almost half of that
reported in RGI (30%). Variability in these figures is possibly due to the differences in the mapping techniques,
thereby increasing the risk of systematic error. Moreover, due to the involvement of different analysts in the
latter, the  results may more likely suffer with random errors.
Results from this study reveal an overall deglaciation of the glaciers in the SSB at an annual rate of ~0.1
±0.0004% during the period 1971-2017. This quantum of area loss is comparatively less to the average annual
rate of 0.4% reported in the western Himalaya (Supplementary table S3). However, our results are comparable
with Birajdar et al. (2014), Chand and Sharma (2015) and Patel et al. (2018) and differ considerably with other
studies in the western Himalayas (Supplementary table S3). Period wise deglaciation varied from 0.1 ±0.0007 to
0.2 ±0.005% $a^{-1}$ during 1971-2000 and 2000-2017, respectively. This result is in line with the recent findings by
Maurer et al. (2019), who suggest a higher average mass loss post 2000 (-0.43 m w.e.$a^{-1}$), which is almost
double the rate reported during 1975-2000 (-0.22 m w.e.$a^{-1}$) for the entire Himalaya.
Comparing the deglaciation rates of the glaciers within the western Himalayan region reveals considerable
heterogeneity therein (Supplementary table S3). It is observed that the Karakoram Himalayan glaciers, in
particular had been losing area till 2000 at an average rate of 0.09% $a^{-1}$, with an increase in area thereafter by
~0.05%$a^{-1}$ (Liu et al., 2006; Minora et al., 2013; Bhambri et al., 2013). However, glaciers in the GHR and Trans
Himalayan range have been deglaciating with higher average annual rate of 0.4 and 0.6%$a^{-1}$, respectively during
the period 1962-2016 (Kulkarni et al., 2007; Kulkarni et al., 2011; Rai et al., 2013; Chand and Sharma, 2015;
Mir et al., 2017; Schmidt and Nusser, 2017; Chudley et al., 2017; Patel et al., 2018; Das and Sharma, 2018). In
contrast to these studies, deglaciation rates in SSB, which comprises of glaciers in GHR as well as LR have
varied from 0.1%$a^{-1}$ (GHR) to 0.2%$a^{-1}$ (LR) (present study). These results evidently depict that the response of
the SSB glaciers is transitional between the Karakoram Himalayan and GHR glaciers. Period wise area loss of
the glaciers in the Himalayan region suggest maximum average deglaciation of eastern (0.49%/yr), followed by
central (0.36%/yr) and western (0.35%/yr) Himalayan glaciers before 2000. Contrarily, after 2000, the central
Himalayan glaciers deglaciated at the maximum rate (0.52%/yr) followed by western (0.46%/yr) and eastern
(0.44%/yr) Himalayan glaciers (Fig.8). Though these rates reflect the possible trend of deglaciation in the
Himalayan terrain, however, any conclusion drawn would be biased due to insufficient data, particularly in
eastern and central Himalaya.

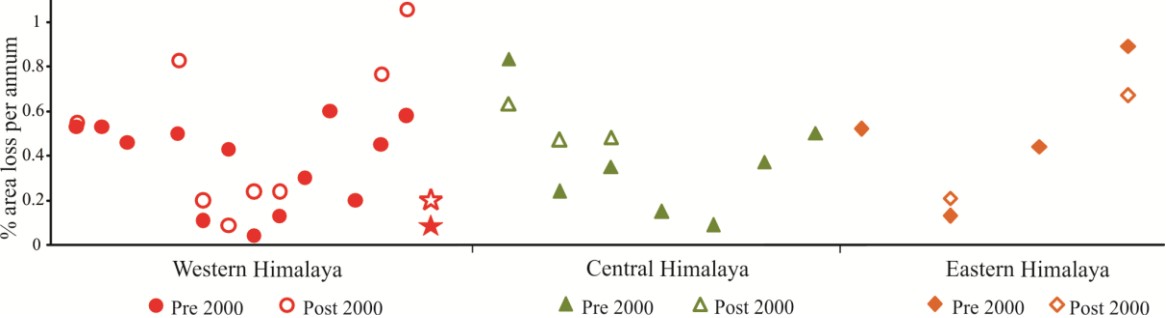

Figure 8: Annual rate of percentage area loss of glaciers in three major sections of Himalaya before and after
2000. Details of the same have been mentioned in Table S3 of Supplementary sheet. Results from the present
study have been star marked in the western Himalaya.

In this study, we found an overall average retreat rate of 4.3 ±1.02 ma$^{-1}$ during the period 1971-2017. However,
the average retreat rates of seven glaciers in the SSB, reported by Kamp et al., (2011) is found to be nearly twice
(24 ma$^{-1}$) of that found in this study (10 ma$^{-1}$). The comparatively higher retreat rates in the former might be due
to the consideration of different time frames. The average retreat rates in other basins of the western Himalaya is
also found to be higher (7.8 ma$^{-1}$) in the Doda valley (Shukla and Qadir, 2016), 8.4 ma$^{-1}$ in Liddar valley
(Murtaza and Romshoo, 2015), 15.5 ma$^{-1}$ in the Chandra-Bhaga basin (Pandey and Venkataraman, 2013) and 19
ma$^{-1}$ in the Baspa basin (Mir et al., 2017). These results show lower average retreat rate of the glaciers in the
SSB as compared to the other studies in the western Himalaya.
The observed average retreat rates during 2000-2017 (4.6 ±1.02 ma$^{-1}$) is found to be nearly twice of that, noted
during 1971-2000 (2 ±1.7 ma$^{-1}$). Similar higher retreat rates post 2000 have been reported in the Tista basin
(Raina, 2009), Doda valley (Shukla and Qadir, 2016), Chandra Bhaga basin (Pandey and Venkataraman, 2013)
and Zanskar basin (Pandey et al., 2011). However, these studies may not sufficiently draw a generalized picture
of glacier recession in the Himalayan region.

**5.2 Spatio-temporal variability in the climate data**
Climatic fluctuations play a crucial role in understanding glacier variability. In this regard, CRU-TS 4.02 dataset
helped in delineating the long term fluctuations in the temperature and precipitation records.
**5.2.1 Basin-wide climate variability**
During an entire duration of 116 years, i.e. from 1901-2017, maximum mean annual temperature is observed in
2016 (3.23 °C) and minimum during 1957 (-0.51 °C). Mean annual temperature shows an almost uniform trend
till 1996, with a pronounced rise thereafter till 2005/06 period (Fig. 3a;b;c). The globally averaged combined
land and ocean surface temperature data of 1983-2012 period are considered as the warmest 30-year period in
the last 1400 years (IPCC, 2013). This unprecedented rate of warming primarily attributed to the rapid scale of
industrialization, increase in regional population and anthropogenic activities prevalent during this time period
(Bajracharya et al., 2008; IPCC, 2013). Thus, one of the probable reason for this sudden increase in temperature
pattern is possibly due to the greenhouse effect from enhanced emission of black carbon in this region (by 61%)
from 1991-2001. Evidences of incessant increase in temperature during 1990s has also been observed (through
chronology of Himalayan Pine) from the contemporaneous surge in tree growth rate (Singh and Yadav 2000). In
fact, 50% of the years since 1970 have experienced considerably high solar irradiance and warm phases of
ENSO, which is possibly one of the reasons for the considerable rise in temperature throughout the Himalaya
(Shekhar et al., 2017). Maximum mean annual precipitation is noted during 2015 (615 mm) and minimum
during 1946 (244 mm). However, the mean annual precipitation followed a similar trend till 1946 with an
increasing thereafter (Fig. 3a;b;c). Besides these general trends in temperature and precipitation,an overall
absolute increase in the mean annual temperature ($T_{max}$ & $T_{min}$) and precipitation data have been noted as 0.77
°C (0.25 °C & 1.3 °C) and 158 mm, respectively during the period 1901-2017. These observations suggest an
enhanced increase in $T_{min}$ by nearly 5 times as compared to the $T_{max}$ alongwith a simultaneous increase in the
precipitation during the period 1901-2017.
Seasonal variations reveal monthly mean temperature and precipitation of 6.7 °C and 1071 mm during summer
(Apr-Oct) and -6.9 °C and 890 mm during winter (Nov-Mar) recorded during 1901-2017 period. Maximum
monthly mean temperature and precipitation have been observed in July (11.8 °C/ 50.4 mm) and August (11.4
°C/ 52 mm) during the period 1901-2017, suggesting them to be the warmest and wettest months. While,
January is noted to be the coldest (-10.4 °C) and November (10.3 mm) to be the driest months in the duration of
116 years (Fig. 3d;e;f). Summer/ winter mean annual temperature and precipitation have increased significantly
by an average 0.74/ 1.28 °C and  85/ 72 mm, respectively during the period 1901-2017. These values reveal a
relatively higher rise in winter average temperature in contrast to the summer. However, enhanced increase in
$T_{min}$ (1.8°C) during winter and $T_{max}$ (0.78°C) during summer have also been observed during the 1901-2017 time
period. The relatively higher rise in the winter temperature (particularly $T_{min}$) and precipitation possibly suggest
that the form of precipitation might have changed from solid to liquid during this particular time span. Similar
increase in the winter temperature have also been reported from the NW Himalaya during the 20[th] century
(Bhutiyani et al., 2007).
In contrast to the long-term climate trends, we have also analyzed the climate data for the study period, i.e.,
1971-2017. An overall increase in the average temperature (0.3°C), $T_{max}$ (0.45°C) $T_{min}$ (1.02°C) and
precipitation by 213 mm is observed. Meanwhile, an enhanced increase in winter $T_{min}$ (1.7°C) and summer $T_{max}$
(0.45°C) are observed. These findings aptly indicate the important role of winter $T_{min}$ and summer $T_{max}$ in the
SSB.
**5.2.2 Local climate variability**
Apart from these generalized climatic variations, grid-wise analysis of the meteorological parameters reveal
existence of local climate variability within the sub-basin (Fig. 3; 9).

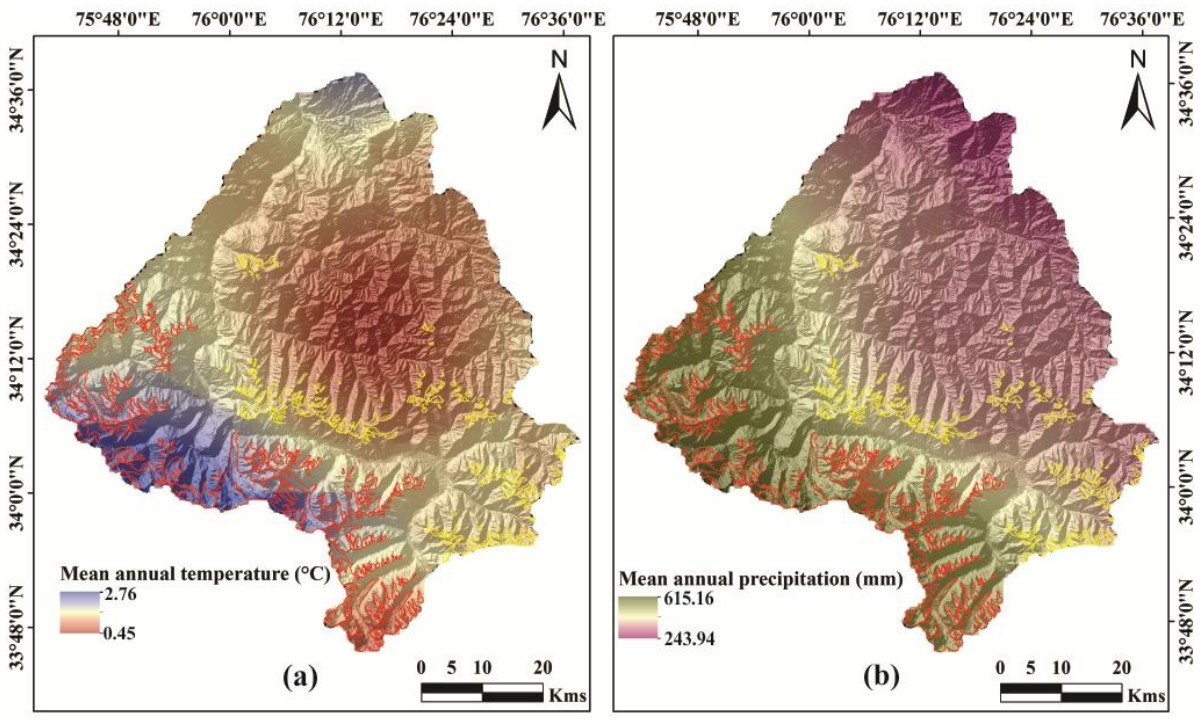


Figure 9: Spatial variation in meteorological data recorded for 15 grids in the SSB during the period 1901-2017.
Map showing the long term mean annual (a) temperature (°C) and (b) precipitation (mm) data within the sub-
basin suggesting the existence of significant local climate variability in the region. Glacier boundaries are shown
as: GHR (red) and LR (yellow).

Observations indicate that the glaciers covered in grid 4 have been experiencing a warmer climatic regimes with
the maximum annual mean temperature of 1.69 °C as compared to the other glaciers in the region (grid 2 = 1.4
°C, grid 5 = 0.74 °C, grid 1 = 0.65 °C and grid 3 = 0.45 °C). Spatial variability in annual mean precipitation data
reveal that grid 2 (448 mm) & grid 1 (442 mm) experienced wetter climate as compared to grid 4 (383 mm),
grid 3 (373 mm) and minimum in grid 5 (318 mm). These observations suggest that GHR glaciers have been
experiencing a warmer and wetter climate (1.03 °C/ 445 mm) as compared to the LR glaciers (0.96 °C/ 358 mm)
(Fig. 3e; f). These observations clearly show that local climate variability does exist in the basin for the entire
duration of 116 years (Fig. 9).

**5.3 Glacier changes: Impact of climatic and other plausible factors**

The alterations in the climatic conditions, discussed in Sect. 5.2, would in turn, influence the glacier parameters,
however varying with time. This section correlates the climatic and other factors (elevation range, regional
hypsometry, slope, aspect and proglacial lakes) with the variations in the glacier parameters.

**5.3.1 Impact of climatic factors**

An overall degenerating pattern of the glaciers in the SSB is observed during the period 1971-2017, with
deglaciation of 32 $\pm$9 km$^2$ (6 $\pm$0.02%). In the same duration, the glaciers have also retreated by an average 199
$\pm$46.9 m (retreat rate: 4.3 $\pm$1.02 ma$^{-1}$) alongwith an increase in the debris cover by ~62%. The observed overall
degeneration of the glaciers have possibly resulted due to the warming of climatic conditions during this
particular time frame. The conspicuous degeneration of these glaciers might have led to an increased melting of
the glacier surface, which in turn would have unveiled the englacial debris cover and increased its coverage in
the ablation zone (Shukla et al., 2009; Scherler et al., 2011). An enhanced degeneration of the glaciers have been
noted during 2000-2017 (0.85 $\pm$0.005 km$^2$a$^{-1}$) than 1971-2000 (0.59 $\pm$0.005 km$^2$a$^{-1}$). Also, nearly 12 glaciers
have shown disintegration into glacierets after 2000. These observations may be attributed to the relatively
higher annual mean temperature (1.68 °C) during the former as compared to the period 1971-2000 (0.89 °C).
Concomitant to the maximum glacier degeneration during the period 2000-2017, debris cover extent has also
increased more (24%) as compared to 1971-2000 (16%). The enhanced degeneration of the glaciers during
2000-2017 might have facilitated an increase in the distribution of supraglacial debris cover. A transition from
CGs to PDGs has also been noticed which resulted due to increase in the debris cover percentage over nearly 99
glaciers. The conversion from PDGs to HDGs (39) and from CGs to HDGs (2) has also occurred. Also, most of
these transitions have occured during 2000-2017, which confirms the maximum degeneration of the glaciers
during this particular period.
It is observed in our study that smaller glaciers have deglaciated more (4.13%) than the medium (1.08%) and
larger (1.03%) sized glaciers during the period 1971-2017 (Supplementary figure S2). This result depicts an
enhanced sensitivity of the smaller glaciers towards the climate change (Bhambri et al., 2011; Basnett et al.,
2013; Ali et al., 2017). A similar pattern of glacier degeneration is noted during 1971-2000, with smaller
glaciers deglaciating more (5%) as compared to the medium sized (3%) and larger (1%) ones. However during
2000-2017, medium glaciers showed slightly greater degeneration (3.9%) as compared to the smaller (3.7%)
followed by larger ones (1.5%). We have also observed maximum length change for smaller glaciers (8%) in
comparison to medium (5%) and large glaciers (3%). These results indicate that the snout retreats are commonly
associated with small and medium sized glaciers (Mayewski et al., 1980).
Temporal and spatial variations in SLAs are an indicator of ELAs, which in turn provide direct evidences
related to the change in climatic conditions (Hanshaw and Bookhagen, 2014). SLAs are amongst the dynamic
glacier parameters that alters seasonally and annually, indicating their direct dependency towards the climatic
factors such as temperature and precipitation. In the present study, the mean SLA has gone up by an average 22
$\pm60$ m during the period 1977-2017. This rise in SLA is synchronous with the increase in mean annual
temperature by 0.43°C. Moreover, the maximum rise in SLA during 1994-2000 is contemporaneous with the
rise of temperature by 0.64 °C during this time period.
Further, in order to understand the regional heterogeneity in glacier response within the sub-basin, parameters of
the GHR and LR glaciers are analyzed separately at four different time periods and correlated with the climatic
variables. It is found that the LR glaciers have deglaciated more (7.2%) as compared to the GHR glaciers
(5.9%). Similarly, more debris cover is found to have accumulated over the LR (73%) glaciers as compared to
the GHR (59%) glaciers during 1971-2017. This result shows that the relatively cleaner (LR) glaciers tend to
deglaciate more alongwith accumulation of more debris as compared to the debris and partially debris covered
glaciers (GHR glaciers) (Bolch et al., 2008; Scherler et al., 2011). Moreover, increase in  mean annual
temperature in the LR (0.3°C) is slightly greater than in GHR (0.25°C) during the period 1971-2017, thus
exhibiting a positive correlation with deglaciation and debris cover distribution in these regions. We also
observed that the glacier area, length and debris cover extent of the LR glaciers show a good correlation with
winter $T_{min}$ and average precipitation as compared to the GHR glaciers (Table 3). This shows that both
temperature as well as precipitation influence the degeneration of the glaciers and in turn affects the supraglacial
debris cover. It is believed that winter precipitation has a prime control on accumulation of snow on the glaciers,
hence acts as an essential determinant of glacier health (Mir et al., 2017). Also, the negative correlation of
glacier area with precipitation in this study possibly indicate the major role of increased winter temperature and
precipitation, which might have decreased the accumulation of snow, thereby decreasing the overall glacier area.
The average SLA for LR glaciers is observed to be higher as compared to the GHR glaciers. However, a
relatively higher rise in SLA is observed for GHR in contrast to the LR glaciers. Also, the mean SLA of the
GHR glaciers shows a good positive correlation with summer $T_{max}$ as compared to the LR glaciers, while a
negative correlation with precipitation in the respective year (Table 3). Considering these observations, it
appears that a general rise in SLA can be attributed to regional climatic warming while that of individual SLA
variation in glaciers may be related to their unique topography (Shukla and Qadir, 2016).
From this analysis, it is quite evident that climatic factors directly influence the glacier response. Also, summer
$T_{max}$ have a stronger control over SLA, while glacier area,length and debris cover are predominantly controlled
by the winter $T_{min}$ in the sub-basin.

Table 3: Coefficients of determination (r) between respective meteorological (temperature and precipitation)
data and observed glacier parameters in the Greater Himalayan Range (GHR) and Ladakh Range (LR) at 90%
confidence.Tavg,Tmin and Tmax are montly mean, monthly mean minimum, monthly mean maximum
temperatures and Pptismontly mean precipitation during different point in time (1971,1994, 2000 and 2017)

| Major Mountain Ranges | Glacier Parameters | Climate Variables | | | |
|---|---|---|---|---|---|
| | | Tavg | Tmin | Tmax | Ppt |
| GHR | Area | -0.826 | **-0.897** | -0.347 | -0.670 |
| | Length | -0.908 | **-0.926** | -0.345 | -0.719 |
| | Debris cover | 0.842 | **0.847** | 0.434 | 0.593 |
| | SLA | 0.725 | 0.209 | **0.725** | -0.315 |
| LR | Area | -0.900 | **-0.942** | -0.568 | -0.779 |
| | Length | -0.909 | **-0.939** | -0.569 | -0.778 |
| | Debris cover | 0.929 | **0.907** | 0.595 | 0.719 |
| | SLA | 0.658 | 0.395 | **0.658** | -0.505 |

**5.3.2 Impact of other factors**
In addition to the climate variables, other factors such as hypsometry, maximum elevation, altitude range, slope,
aspect and proglacial lakes also influence the response of individual glacier.
Glacier hypsometry is a measure of mass distribution over varying altitudes. It is affected by the mean SLA of
the glaciers to a greater extent, as it is considered that if a large portion of the glacier has elevation equivalent to
SLA, then even a slight alteration in SLA might significantly change the ablation and accumulation zones
(Rivera et al., 2011; Garg et al., 2017b).
In this study, we observed that GHR and LR glaciers have nearly 45% and 10% of their area at an elevation
similar to SLA. This suggests that GHR glaciers are more susceptible to retreat as compared to the LR glaciers,
as a larger portion of the former belongs to the SLA. Moreover, the hypsometric distribution of glacier area in
the GHR and LR of the SSB reveals maximum area change post 2000 (Fig.6b). In this regard, while GHR
glaciers have undergone relatively higher area loss (21%) at lower elevation (3800-4200 masl), the LR glaciers
lost maximum area (30%) at much higher elevation (5600-5900 masl) ranges (Fig.6b). Besides, a significant
area loss has also been observed for both GHR (6%) and LR (7%) glaciers at their mean elevations post 2000
(Fig.6b).
Elevation plays an important role in understanding the accumulation pattern at higher and ablation in the lower
altitudes. The general perception is that the glaciers situated at relatively higher elevation are subjected to
greater amount of precipitation and hence are susceptible to less deglaciation or even mass gain (Pandey and
Venkataraman, 2013). Similarly, we have also noticed that the glaciers extending to comparatively higher
maximum elevation experience minimum retreat (10%) and exhibit higher percentagedeglaciation (33%) as
compared to the glaciers having lower maximum elevation (retreat:15% & deglaciation: 20%) (Fig.10a).

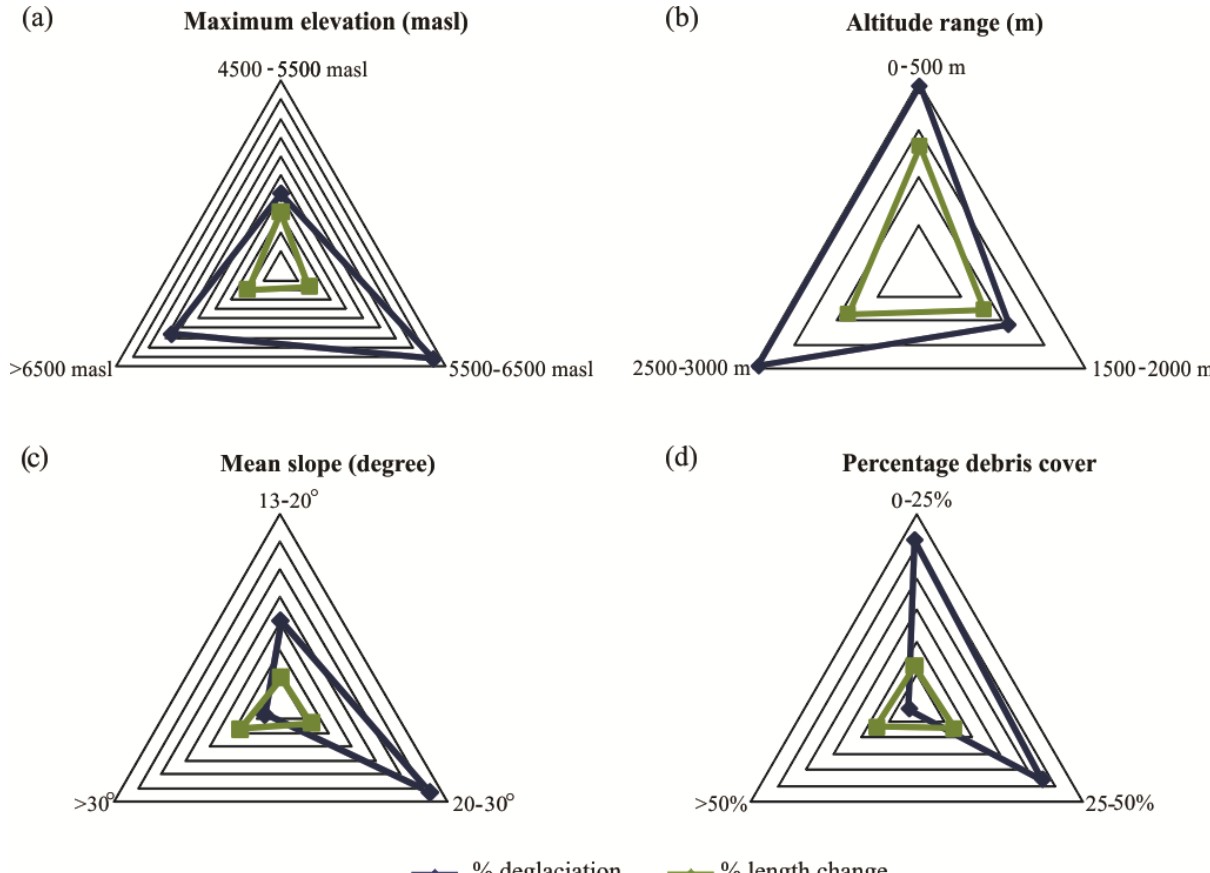

Figure 10: Differential degeneration of the glaciers during the period 1971-2017 with variability in non-climatic factors. (a) Percentage deglaciation and length change of the glaciers at different ranges of maximum elevation, (b) altitude range, (c) mean slope and (d) percentage debris cover.

Moreover, our study shows that the glaciers having lower altitude range have retreated and deglaciated more (13% & 20%, respectively) as compared to the counterparts (Fig.10b). These observations indicate that glaciers which possess higher maximum elevation and altitudinal range are subjected to less retreat and undergo greater deglaciation.

Slope is another important factor which has a major role in the sustenance of the glacier as accumulation of ice is facilitated by a gentler bedrock topography (DeBeer and Sharp, 2009; Patel et al., 2018). It is observed that glaciers having steep slopes (30-40°) have retreated more (17%), however with minimum deglaciation (7%) during the period 1971-2017 (Fig.10c). Similar results with steeper glaciers exhibiting minimum deglaciation have been reported in the Parbati, Chandra and Miyar basins (Venkatesh et al., 2012; Patel et al., 2018). However, it differs with Pandey and Venkataraman (2013) and Garg et al., (2017b), likely due to the differing average size: 25 ±33.78 and 17 ±33.2 km$^2$ (present study: 2 ±5.7 km$^2$) and slope: 5-20° and 12-26° (present study: 13-41°), respectively, of glaciers used in these studies.

Presence of supraglacial debris cover influences the glacier processes. Depending on thickness, debris cover may either enhance or retard the ablation process (Scherler et al., 2011). In this study, we observed that clean glaciers have undergone maximum deglaciation (52%) as compared to the partially (46%) and heavily debris covered glaciers (2%). However, they all have retreated almost similarly (12 to 14%), with slightly higher retreat of partially debris covered glaciers (Fig.10d). Aspect/ orientation of glaciers provide information

regarding the duration for which they are exposed to the incoming solar radiation. Since, the south facing
glaciers are subjected to longer duration of exposure to the solar radiations as compared to the north facing
glaciers, therefore, are prone to greater deglaciation and retreat (Deota et al., 2011). Here, it is observed that the
glaciers having northerly aspect (north, north-east, north-west) have undergone maximum deglaciation as
compared to the counterparts. However, majority (71%) of the glaciers have northerly aspect, so any inferences
drawn in this respect would be biased. It is worthwhile to state that most of the south facing slopes in the basin
are devoid of glaciers but show presence of relict glacier valleys which would have been glaciated in the past.
At present only 48 south facing glaciers (south, south-east, south-west) with an average size of 1 $\pm1.9$ km$^2$ exist
in the SSB.
Similarly, the glacier changes are also influenced by the presence of certain features such as glacial (proglacial
or supraglacial) lakes or differential distribution of supraglacial debris cover. The presence of a proglacial or
supraglacial lakes significantly enhances the rate of glacier degeneration by increasing the melting processes
(Sakai, 2012; Basnett et al., 2013). As per our results, highest average retreat rate (~31 ma$^{-1}$) is observed for
glaciers G-4 (Dulung glacier). Although, it is a debris free glacier, shows the highest retreat rates. Also, a
moraine-dammed lake is observed at the snout of this glacier and has continuously increased its size from 0.15
km$^2$ in 1977 to 0.56 km$^2$ in 2017. This significant increase in the size of moraine-dammed lake has possibly
influenced the enhanced retreat rate of the glacier.
**6 Dataset availability**
Temporal inventory data for glaciers of Suru sub-basin, western Himalaya is available at
https://doi.pangaea.de/10.1594/PANGAEA.904131 (Shukla et al., 2019).

**7 Conclusions**
The major inferences drawn from the study include:
1. The sub-basin comprised of 252 glaciers, covering an area of 481.32 $\pm3.41$ km$^2$ (11% of the glacierized area)
in 2017. Major disintegration of the glaciers occurred after 2000, with breakdown of 12 glaciers into glacierets.
Small (47%) and clean (43%) glaciers cover maximum glacierized area of the sub-basin. Topographic
parameters reveal that majority of the glaciers are north facing and the mean elevation and slope of the glaciers
are 5134.8 $\pm225$ masl and 24.8 $\pm5.8°$, respectively.

2. Variability in glacier parameters reveal an overall degeneration of the glaciers during the period 1971-2017,
with deglaciation of approximately 0.13 $\pm0.0004\%$ a$^{-1}$ alongwith an increase in the debris cover by 37 $\pm0.002$
km$^2$ (~62%). Meanwhile, the glaciers have shown an average retreat rate of nearly 4.3 $\pm1.02$ ma$^{-1}$ with SLA
exhibiting an overall rise by an average 22 $\pm60$ m.
3. Long-term meteorological records during the period 1901-2017 exhibit an overall increase in the temperature
($T_{min}$: 1.3°C, $T_{max}$: 0.25°C, $T_{avg}$: 0.77°C) and precipitation (158 mm) trends. Both temperature and precipitation
gradients influence the changes in glacier parameters, however, winter $T_{min}$ strongly influencing the glacier area,
length and debris cover while summer $T_{max}$ controlling the SLA. Spatial patterns in change of climate
parameters reveal existence of local climate variability in the sub-basin, with progressively warmer (1.03°C) and
wetter (445 mm) climatic regime for glaciers hosted in the GHR as compared to the LR (0.96°C/ 358 mm).
4. The inherent local climate variability in the sub-basin has influenced the behavior of the glaciers in the GHR
and LR. It has been observed that LR glaciers have been shrinking faster (area loss: 7%) and accumulating more
debris cover (debris increase: 73%) as compared to the GHR glaciers (6% and 59%) during the period 1971-
2017. The GHR glaciers have, however, experienced greater rise in SLA (220 ±*121* m) in comparison to the LR
ones (*91 ±56* m) during the period 1977-2000, with a decrease thereafter.

Results presented here show the transitional response of the glaciers in the SSB between the Karakoram
Himalayan and GHR glaciers. The study also confirm the possible influence of factors other than climate such
as glacier size, regional hypsometry, elevation range, slope, aspect and presence of proglacial lakes in the
observed heterogenous response of the glaciers. Therefore, these factors need to be accounted for in more details
in future for complete understanding of the observed glacier changes and response**.**

**Team list**
1. Aparna Shukla
2. Siddhi Garg
3. Manish Mehta
4. Vinit Kumar
5. Uma Kant Shukla

**Author contribution**
A.S. and S.G. conceived the idea and led the writing of manuscript. A.S. structured the study. S.G. performed
the temporal analysis of the data. M.M. and V.K. helped in the field investigation of the glaciers. All the authors
helped in interpretation of results and contributed towards the final form of the manuscript.

**Competing interests**
The authors declare that they have no conflict of interest.

**Acknowledgements**
Authors are grateful to the Director, Wadia Institute of Himalayan Geology, Dehradun for providing all the
research facilities and support for successful completion of this work. We wish to convey our sincere thanks to
the anonymous reviewers for detailed reviews and constructive comments, which greatly helped to improve the
previous version of the manuscript. We are thankfull to Prasad Gogineni (Handing Topical Editor) and Jens
Klump (Handing Chief Editor) for their thoughtful suggestions on the manuscript. Also, we appreciate the
efforts of the entire Editorial team of Earth System Science Data (ESSD) for timely processing of the article.
Aparna Shukla acknowledges the Secretary, Ministry of Earth Science (MoES), New Delhi, India, for providing
requisit support.

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
