# Peer review of "Temporal inventory of glaciers in the Suru sub-basin, western"

_Earth System Science Data, 2019_

## Referee Comment (RC1) · Anonymous Referee #1 · 25 Sep 2019

Paper entitled 'Temporal inventory of glaciers in the Suru sub-basin, western Himalaya: Impacts of the regional climate variability' by Shukla et. al. presents an analysis of changes in the 252 glaciers of the Suru sub-basin from 1971 to 2017 using remote sensing data. Paper is well written, interesting and present an insight in the variation of the glaciers of different Himalayan ranges i.e. Great Himalaya and Ladakh range. However there are number of flaws in the manuscript, particularly related to data presentation and interpretation. These flaws need to be addressed. Manuscript needs a major revision.

1. Long-term climate data presentation and analysis needs attention. Page 8, Line

183; Mean precipitation of the SSB for the period 1901-2017 has been 393+-76 mm. However, If we see plots in Figure 3 (d), 3 (c) and 3 (f), monthly mean precipitation for the same period are quite high indicating high precipitation during the same period.

2. Figure 3(a), 3(b) and 3(c) shows continuous increase in the temperature during the period 1995-96 onwards till 2005-06. It shows sudden change in temperature pattern. The reason for the sudden shift in temperature pattern should be discussed. It will be interesting to see the temperature pattern of the IMD recorded data at Leh or any other in-situ recorded data in the study region during the same period.

3. A comparison of the CRU data with in-situ data (temperature and precipitation) in the study region will provide information about the biases in the CRU data.

4. Page 13, Line 339; How authors will explain the mean slope variation of 16.2°+-71° to 41° +- 66° .

5. Figure 4(a) Frequency distribution histogram depicting maximum frequency in the perent area change between 0.52 - 0.97. How it is conclude that majority of the glaciers have undergone an area loss of 3.3%.

6. Figure 5; Majority of the glaciers have undergone length change of 5% is not seen in the frequency distribution histogram.

7. What could be the possible reasons of decrease in SLA in LR glaciers despite of increase in temperature and retreat in glacier length in the region?

8. Page 16, Line 405; there is a large difference in the number of glaciers reported in the sub basin by earlier researchers and reported in the present paper. It needs discussion and possible reasons. Is there any difference in defining a glacier?

9. Page 18, Line 462; statement 'However a sudden decrease in the precipitation anomaly is observed in the year 2016 with an increase thereafter', it is not clear to me that Figure 3(a),(b) and (c) are showing 'precipitation' or 'precipitation anomaly'? Year 2016 is missing in the Figure.

10. Page 18, Line 462-463; statement regarding mean annual precipitation is not clear if I look at Figure.

11. Page 18, Line 463-464; 'temperature and precipitation anomaly' not understood.

12. It is advised to draw a trend line for temperature and precipitation variation in Figure 3.

13. Page 18, Line 466; 'Percentage increase in the average, maximum and minimum temperature observed to be 99, 12 and 17%', generally temperature variation is not shown in percentage. I will give an example, if mean temperature varies from 0.1°C to 0.2°C for one year and next year it drops to 0.1°C again, should one conclude that temperature variation was 100% increasing for the first year and 100% decreasing for next year. Statement will be misleading, since the temperature variation was minimal. If the unit of temperature changes from °C to K, then still the statement will hold good? It is advised not to represent temperature variation in % throughout the manuscript.

---

## Referee Comment (RC2) · Anonymous Referee #2 · 17 Nov 2019

The study by Shukla et al. entitled, "Temporal inventory of glaciers in the Suru sub-basin, western Himalaya ....." provides very useful data sets glaciers in the Suru Sub Basin in Western Himalaya that are very useful for better understanding the status and fate of the glaciers in the Western Himalaya. The data and manuscript quality is good, except that it would require a major revision to make it in the framework of data paper. Currently, larger focus is on the scientific implications of the data, which is not focus of the journal. While authors have also followed standard methods to process and analyse the data, the methods are not unique.

Overall, large amount of digitization work has been done for this study. However, the

[Figure]

Suru basin is a small sub-basin of the Indus river basin, with only 11% of its area is covered with glaciers. So the authors need to substantially revise the manuscript to be useful as a regional representative of Western Himalayan glaciers. Considering the unique scope of the journal, it would therefore, require that the authors to incorporate similar dataset from other distinct basins of Upper Indus Basin to make it more regionally relevant.

Few specific comments are:

1. Unlike the Karakoram , the Ladakh Range is not a well known nomenclature. Chudley et al., (2017) have used the Karakoram and Ladakh range, not differentiated about Karakoram and GHR. Mir et al. (2018) have represented it as a part of the GHR. It is therefore, important to define/clarify the same.

2. The accuracy of CRU-TS data is not analysed independently. It is critical as the Fig. 3 data looks bit unrealistic. The temperature data indicate dramatic changes after 1990, which needs to be confirmed. Since India Met Department has long term station data in this region as well as gridded data (http://www.imdpune.gov.in/Clim_Pred_LRF_New/Grided_Data_Download.html), it is critical to check the data consistency and conduct error statistics.

3. Considering the large uncertainty involved in LandsatMSS data, it s important to mention the inherent uncertainties while interpreting the temporal variability. Table 1: include the Scene ID for clarity.

4. Lines 236- 240: The procedures used for determining the glacier boundaries are apparently manual digitization. While this is reasonable to undertake manual processing in such complicated areas, it also necessitates a study of uncertainty estimations in such manual work. Authors may also undertake repeatability tests with different analysts to determine repeatability.

5. Lines 272 – 300: The uncertainty assessment is biased with the very limited field

validation on only one glacier for a very limited time frame. One issue that needs to be addressed is the reliability of ground truth data when different types of data were used through the nearly 50 years time period.

6. Please discuss why the projective transformation was required for the satellite data sets other than Corona?

7. Line 328- 330: Categorization of glaciers - is there a scientific standard for categorizing the glaciers in the different categories or was more based on the author's selectivity? Check DeBeer and Sharp (2009, Journal of Glaciology). Since the data descriptions needs to be internationally consistent, may revise.

8. Statistical significance could be included to explain the effect spatial characteristics (size, aspect, debris cover) or any difference spatial control over LR and GHR.

---

## Author Comment (AC1) · 10 Dec 2019

Authors sincerely thank the editorial team for timely processing of the article and the referee for his/her valuable comments and suggestions on our manuscript. Please find the detailed specific responses to the referee's comments below. Revisions to the manuscript are mentioned in track change mode in the revised manuscript and also appended at the end of this document. Regards. Aparna Shukla.

Referee # 1:

Comment 1: Long-term climate data presentation and analysis needs attention. Page

[Figure]

8, Line 183; Mean precipitation of the SSB for the period 1901-2017 has been 393 ±76 mm. However, if we see plots in figure 3(d), 3(e) and 3(f), monthly mean precipitation for the same period are quite high indicating high precipitation during the same period. Response 1: Thanks for pointing out. The annual average precipitation for the Suru sub-basin amounts to 393 ±76 mm during the period 1901-2017. However, the monthly mean precipitation values during the same period had been overestimated due to computational error. This error was introduced due to the variance in formats available for the CRU-TS derived precipitation data and hence was mistaken with the other format (mm/day). The error has now been rectified in the revised manuscript (Page 8; Figure 3d, 3e & 3f). The revised figures (3d, 3e, 3f) show monthly mean precipitation (Jan-Dec) variations of 33 ±14 mm/month in the entire Suru sub-basin, while 37 ±15 mm/month and 30 ±12 mm/month in the GHR and LR, respectively during the period 1901-2017.

Figure 3 (revised manuscript): Annual and seasonal variability in the climate data for the period 1901-2017. (a), (b) and (c) 5 year moving average of the mean annual precipitation (mm) and temperature (°C) recorded for 5 grids covering the glaciers in the entire Suru sub-basin (SSB), Greater Himalayan range (GHR) and Ladakh range (LR) (sub-regions), respectively during the period 1901-2017. The light and dark grey colored lines depict the respective trend lines for precipitation and temperature conditions during the period 1901-2017. (d), (e) and (f) Monthly mean precipitation and temperature data for the entire SSB, GHR and LR (sub-regions), respectively for the time period 1901-2017.

Comment 2: Figure 3(a), 3(b) and 3(c) shows continuous increase in the temperature during the period 1995-96 onwards till 2005-06. It shows sudden change in the temperature pattern. The reason for the sudden shift in temperature pattern should be discussed. It will be interesting to see the temperature pattern of the IMD recorded data at Leh or any other in-situ recorded data in the study region during the same period. Response 2: Agreed. The mean annual temperature depicted in figure 3(a), 3(b) and

3(c) shows an overall increase of 0.71°C, 0.72°C, 0.71°C in the Suru sub-basin, GHR and LR, respectively, during 1995/96 till 2005/06 period as mentioned by the referee. The globally averaged combined land and ocean surface temperature data of 1983-2012 period is considered as the warmest 30-year period in the last 1400 years (IPCC, 2013). This unprecedented rate of warming has been primarily attributed to the rapid scale of industrialization, increase in regional population and anthropogenic activities prevalent during this time period (Bajracharya et al., 2008; IPCC, 2013). Thus, one of the probable reason for this sudden increment in temperature pattern is possibly due to the greenhouse effect from enhanced emission of black carbon in this region (by 61%) from 1991-2001 (Sahu et al., 2008). Evidences of incessant increase in temperature during 1990s have also been observed (through chronology of Himalayan Pine) from the contemporaneous surge in tree growth rate (Singh and Yadav 2000). In fact, 50% of the years since 1970 have experienced considerably high solar irradiance and warm phases of ENSO, which is possibly one of the reasons for the considerable rise in temperature throughout the Himalaya (Shekhar et al., 2017). The same has now been discussed in the revised manuscript as suggested (Page 17, lines 470-480). Due to the unavailability of in-situ climate dataset for the Suru sub-basin, station data is obtained from nearest stations of Kargil and Leh and compared with the CRU-TS derived data for the entire Suru sub-basin during 1901-2002 period.

Figure R1: Mean annual temperature and precipitation patterns of CRU-TS derived gridded data and IMD recorded station at different locations.

The mean annual temperature pattern of Suru sub-basin shows a near decreasing trend till 1936, with an increase thereafter. Similar trends have been observed for Kargil and Leh, despite their distant location from the Suru sub-basin (areal distance of Kargil and Leh is ∼63 and 126 km, respectively from the centre of Suru sub-basin). However, it is noteworthy to mention that all the locations had attained maximum mean annual temperature in 1999 (Suru: 2.02°C; Kargil: 6.84°C; Leh: -0.5°C). Indeed, these results are interesting and we observe an almost similar trend in all the cases (Figure
R1), with an accelerated warming post 1995/96. However, the magnitude varies, with longterm mean annual temperature of 0.9, 5.5 and -2.04 °C observed in Suru sub-basin, Kargil and Leh, respectively (Figure R1). While the change (increase) in mean annual temperature observed during the same period, i.e., 1901-2002 is found to be 0.34, 0.13 and 0.44 °C in Suru sub-basin, Kargil and Leh, respectively. The possible reason for this difference in their magnitudes could possibly be attributed to their distinct geographical locations and difference in their nature, with former being point, while latter being the interpolated gridded data.

Comment 3: A comparison of the CRU data with in-situ (temperature and precipitation) in the study region will provide information about the biases in the CRU data. Response 3: Agreed. Due to the unavailability of meteorological observatories in the Suru sub-basin, station data is obtained from nearest available IMD sites, i.e., Kargil and Leh and compared with their respective CRU-TS data (mean annual temperature and precipitation).

Figure R2: Mean annual temperature and precipitation patterns of IMD recorded station data at Kargil and Leh and their respective CRU-TS derived gridded data.

Though varying in magnitude, the climate data obtained from IMD as well as CRU-TS suggest almost similar trends of temperature and precipitation during the period 1901-2002 for both Kargil and Leh (Figure R2). The annual mean temperature/ precipitation have amounted to 5.5°C/589 mm (IMD) and 2.4°C/315 mm (CRU-TS) in Kargil, while -2.04/279 mm (IMD) and -0.09/ 216 mm (CRU-TS) in Leh during the period 1901-2002 (Figure R2).We observed that climatic variables show lower magnitude in case of CRU-TS as compared to the station data from IMD (except CRU-TS derived temperature data recorded for Leh). The possible reason for this difference between CRU-TS and station data can primarily be attributed to the difference in their nature, with former being point, while latter being a gridded data (0.5° latitude and longitude grid cells). This analysis aptly brings out the bias in the CRU TS gridded data. Majorly the comparison shows that though the gridded data correctly bring out the temporal trends in

meteorological data but differ with station data in magnitude (being on lower side than the station estimates). This helps us better appreciate the climate variations in the Suru sub-basin as well, since we learn that the reported temperature and precipitation changes are probably on the lower side of the actual variations. In case the reviewer thinks it's appropriate, this analysis may be incorporated at some suitable place in the manuscript or supplementary data.

Comment 4: Page 13, Line 339; How authors will explain the mean slope variation of $16.2° \pm 71°$ to $41° \pm 66°$? Response 4: Thanks for pointing it. In this study, range of slope was reported initially depicting minimum and maximum variations in the overall data, i.e., in $16.2° \pm 71°$, $16.2°$ was the average minimum slope and $71°$ was the deviation in this minimum slope considering the entire basin. Similarly, in $41° \pm 66°$, $41°$ was the average maximum slope while $66°$ was the deviation in this maximum slope considering the entire basin. However, we now realize this form of data representation misleading. Therefore, we have now mentioned the mean slope of $24 \pm 6°$ and $25 \pm 6°$ in GHR and LR glaciers, respectively. The same has now been incorporated in the revised manuscript (Page: 13, Line: 347). Comment 5: Figure 4(a) Frequency distribution histogram depicting maximum frequency in the percent area change between 0.52-0.97. How it concludes that majority of the glaciers have undergone an area loss of 3.3%.

Response 5: The statement mentioning that the majority of the glaciers have undergone area change of 3.3% was based on mid-point of a legend category (0.8-6%) as shown in the chloropleth map. This was misleading as the categories of percent area change depicted in histogram differed from those shown in the chloropleth map. However, now we have simplified the histogram and the chloropleth map by keeping same divisions (range of percent area change) for both. In the revised Figure 4a, it may be observed that majority of the glaciers have undergone area change of the range 6-12% and same is depicted in the chloropleth.

Figure 4: (a) Percent area loss of the glaciers in the SSB during the period 1971-2017.

Frequency distribution histogram depicting that majority of the glaciers have undergone an area loss in the range 6-12%. (b) Hypsometric distribution of glacier area in the GHR and LR regions during the period (I) 1971-2000 and (II) 2000-2017. (A), (B), (C) and (D) insets in (II) shows the significant change in area at different elevation range of the GHR and LR glaciers.

Comment 6: Figure 5; Majority of the glaciers have undergone length change of 5% is not seen in the frequency distribution histogram. Response 6: The statement mentioning that the majority of the glaciers have undergone length change of 5% was based on mid-point of a legend category (0.9-8%) as shown in the chloropleth map. This was misleading as the categories of percent length change depicted in histogram differed from those shown in the chloropleth map. However, now we have simplified the histogram and the chloropleth map by keeping same divisions (range of percent length change) for both. In the revised Figure 5, it may be observed that majority of the glaciers have undergone length change of the range 6-14% and same is depicted in the chloropleth.

Figure 5: Percent length change of the glaciers in the SSB during the period 1971-2017. Frequency distribution histogram showing that majority of the glaciers have undergone length change in the range 6-14%.

Comment 7: What could be the possible reasons of decrease in SLA in LR glaciers despite of increase in temperature and retreat in glacier length in the region? Response 7: Yes, if we simply try to equate the absolute temperature change in LR with the overall SLA and/or length changes observed in this region then the results might seem counter-intuitive. However, such is not the case. While SLA (often used as a reliable proxy for glacier mass balance changes (Guo et al., (2014)) responds directly to the changes in meteorological variables mostly temperature, length changes or retreat are much delayed response of the glaciers towards climate change (Bolch et al., 2012; Paul et al., 2017). Besides, glacier retreat is often strongly influenced by the local snout characteristics and conditions such as presence of proglacial lakes, supraglacial

debris coverage and differential shadowing (Sakai, 2012; Shukla and Qadir, 2016; Garg et al., 2017). For these reasons SLA and retreat trends may not always be in-sync. Coming to the reported increase in temperature in the LR, this increase has been estimated using following formulation which takes into account longterm mean and trends of entire temperature data series in the form of Sen's slope. Change in Temperature and Precipitation=$(\beta*L)/M$ where $\beta$ is Sen's slope estimator, L is length of period and M is the long term mean. Contrary to this, the reported SLA changes are simple difference between the average SLAs of 1977 and 2017. Thus, the SLA changes seem counter-intuitive to the temperature variations and do not correlate well with it. However, if we break this long time frame of 40 years (1977–2017) into shorter time periods then we find that the SLA in LR had been responding excellently to the ongoing temperature changes (Table R1). Also, the SLA and temperature changes have, as expected, high negative correlation with each other (i.e., -0.82). Table RT1: Period wise variations in SLA of the LR glaciers and changes in temperature conditions during the corresponding time interval.

Comment 8: Page 16; Line 405; there is a large difference in the number of glaciers reported in the sub basin by earlier researchers and reported in the present paper. It needs discussion and possible reasons. Is there any difference in defining a glacier? Response 8: Statistics of the year 2000 reveal a total of 240 glaciers in the Suru sub-basin (Page 16; Line: 404 of the original manuscript). This is, though comparable with that reported by Sangewar and Shukla, (2009) i.e. 284, varies drastically from SAC report, (2016) and RGI (2 different analysts) [110 and (514 & 304), respectively].One possible reason could be the difference in methodology adopted for glacier delineation leading to systematic errors (Page: 16; Lines: 410-411 of the original manuscript). Secondly, the involvement of multiple analysts may introduce random errors, as in case of RGI (Page: 16; Lines: 411-412 of the original manuscript). Yes, as already pointed out there is a difference in defining a glacier in these studies, which is yet another plausible reason for introducing the bias in the glacier count. RGI have provided separate glacier id to each polygon in the Sub-basin, which might be the reason for overestimation of

glaciers. While no such information regarding the definition of glacier has been pro-
vided in the SAC report, 2016. However, in this study, the glacierets / tributary glaciers
contributing to the main trunk are considered as a single glacier entity, which is a stan-
dard procedure for assigning the glacier id. The statement was somehow missing from
the original manuscript which would have created the confusion, therefore, now it has
been incorporated in the revised manuscript (Page 10, lines 239-240).

Comment 9: Page 18, Line 462; statement 'However a sudden decrease in the precip-
itation anomaly is observed in the year 2016 with an increase thereafter', it is not clear
to me that Figure 3(a), (b) and (c) are showing 'precipitation' or 'precipitation anomaly'?
Year 2016 is missing in the Figure. Response 9: Figure 3(a) (b) and (c) are showing
'5 year moving average of average annual precipitation'. The statement mentioned in
the original manuscript regarding 'precipitation anomaly' was previously included in the
graphs. However, these graphs were changed (with different mode of representation)
later owing to more information shown by present graphs included in the manuscript
(Figure 3). These lines should have been removed from the text as well. We regret their
inclusion. The vertical bars show 5 year moving average of mean annual precipitation
during the period 1901-2017.

Comment 10: Page 18, Line 462-463; statement regarding mean annual precipitation
is not clear if I look at Figure. Response 10: Similar to Response 9

Comment 11: Page 18, Line 463-464; 'temperature and precipitation anomaly' not
understood. Response 11: Thanks for pointing out. The statement mentioned in the
original manuscript regarding 'temperature and precipitation anomaly' was previously
included in the graphs. However, these graphs were changed (with different mode
of representation) later owing to more information shown by present graphs included
in the manuscript (Figure 3). These lines should have been removed from the text
as well. We regret their inclusion. The statement has now been edited to "Besides
these general trends in mean annual temperature and precipitation, an overall absolute
increase in the mean annual temperature (Tmax & Tmin) and precipitation data have

been noted as 0.77 °C (0.25 °C & 1.3 °C) and 158 mm, respectively during the period 1901-2017" (Revised manuscript; Page 17; lines 482-484).

Comment 12: It is advised to draw a trend line for temperature and precipitation variation in Figure 3. Response 12: Thanks for the suggestion. A trend line for temperature and precipitation variations have now been added in Figure 3 of the revised manuscript.

Comment 13: Page 18, Line 466; 'Percentage increase in the average, maximum and minimum temperature observed to be 99, 12 and 17%', generally temperature variation is not shown in percentage. I will give an example, if mean temperature varies from 0.1°C to 0.2°C for one year and next year it drops to 0.1°C again, should one conclude that temperature variation was 100% increasing for the first year and 100% decreasing for next year. Statement will be misleading, since the temperature variation was minimal. If the unit of temperature changes from °C to K, then still the statement will hold good? It is advised not to represent temperature variation in % throughout the manuscript. Response 13: Agreed. As suggested, we have now reported the temperature and precipitation changes in absolute form rather than in percentage.

REFERENCES Bajracharya, S. R., Mool, P. K., Shrestha, B. R.: Global climate change and melting of Himalayan glaciers. Melting glaciers and rising sea levels: Impacts and implications, Prabha Shastri Ranade (ed), The Icfai's University Press, India, 28–46, 2008. Bolch, T., Kulkarni, A., Kääb, A., Huggel, C., Paul, F., Cogley, J. G., Frey, H., Kargel, J. S., Fujita, K., Scheel, M., Bajracharya, S., and Stoffel, M.: The State and Fate of Himalayan Glaciers, Science, 336, 310–314, https://doi.org/ 10.1126/science.1215828, 2012. Garg, P. K., Shukla, A. and Jasrotia, A. S: Influence of topography on glacier changes in the central Himalaya, India, Global and Planetary change, 155, 196-212, https://doi.org/ 10.1016/j.gloplacha.2017.07.007, 2017. Guo, Z., Wanga, N., Kehrwald, N. M., Mao, R., Wua, H., Wu, Y. and Jiang, X.: Temporal and spatial changes in western Himalayan firn line altitudes from 1998 to 2009, Global and Planetary Change, 118, 97–105,https://doi.org/10.1016/j.gloplacha.2014.03.012, 2014. IPCC. Summary for policymakers. In: Stocker, T. F. et al. (Eds), Climate Change

2013: The Physical Science Basis. Contribution of Working Group III to the Fifth Assessment Report of Intergovernmental Panel on Climate Change. Cambridge University Press, Cambridge and New York, 2013. Paul, F., Bolch, T., Briggs, K., Kääb, A., McMillan, M., McNabb, R., Nagler, T., Nuth, C., Rastner, P., Strozzi, T. and Wuite, J.: Error sources and guidelines for quality assessment of glacier area, elevation change, and velocity products derived from satellite data in the Glaciers_cci project, Remote sensing of Environment, 203, 256-275, https://doi.org/10.1016/j.rse.2017.08.038, 2017. Sakai, A.: Glacial lakes in the Himalayas: A review on formation and Expansion process, Global environmental research, 23-30, 2012. Sangewar, C. V., and S. P. Shukla.: Inventory of the Himalayan Glaciers: A Contribution to the International Hydrological Programme, An Updated Edition. Kolkata: Geological Survey of India (Special Publication 34), IISN: 1:0254–0436, 2009. Shekhar, M., Bhardwaj, A., Singh, S., Ranhotra1, P. S., Bhattacharyya, A., Pal, A. K., Roy, I., Martín-Torres, F. J. and Zorzano, M.P.: Himalayan glaciers experienced significant mass loss during later phases of little ice age, Scientific Reports, 7, 1-14, 2017. Shukla, A. and Qadir, J.: Differential response of glaciers with varying debris cover extent: evidence from changing glacier parameters, International Journal of Remote Sensing, 37, 2453–2479, http://doi.org/10.1080/01431161.2016.1176272, 2016. Singh, J. and Yadav, R. R.: Tree-ring indications of recent glacier fluctuations in Gangotri, western Himalaya, India, Current Science, 79(11), 1598–1601, 2000. Space Application Centre (SAC): Report: Monitoring Snow and Glaciers of Himalayan Region. Space Application Centre, ISRO, Ahmedabad, India, 413 pages, ISBN: 978-93-82760-24-5, 2016.

AUTHOR'S CHANGES IN THE MANUSCRIPT On Page 8; Figure 3 has been updated as suggested by reviewer in Comments 1 and 12.Line stating "The glacierets/ tributary glaciers contributing to the main trunk are considered as single glacier entity" has been added on Page 10; lines: 239-240 according to comment 8.

On page 13, lines: 346-347 have been edited to "Mean slope of the glaciers is 24.8 $\pm$5.8° and varies from 24 $\pm$6° to 25 $\pm$6° in the GHR and LR, respectively" based on

reviewer's comment 4.

Page 13 and 14, lines: 354-355 and figure 4 have been edited as suggest in comment 5.Percentage area loss of the individual glaciers ranges between 0.8 (G-50; Parkachik glacier) - 45 (G-81)%, with majority of the glaciers undergoing an area loss in the range 6-12% during the period 1971-2017 (Fig.4a).

Page 14, lines: 373-374 and figure 5 have been edited as suggest in comment 6. Percentage length change of the glaciers ranges between 0.9 to 47%, with majority of the glaciers retreating in the range 6-14% during the period 1971-2017 (Fig.5).

Page 17, lines: 470-480 stating: Mean annual temperature shows an almost uniform trend till 1996, with a pronounced rise thereafter till 2005/06 period (Fig. 3a; b; c). The globally averaged combined land and ocean surface temperature data of 1983-2012 period is considered as the warmest 30-year period in the last 1400 years (IPCC, 2013). This unprecedented rate of warming has been primarily attributed to the rapid scale of industrialization, increase in regional population and anthropogenic activities prevalent during this time period (Bajracharya et al., 2008; IPCC, 2013). Thus, one of the probable reason for this sudden increment in temperature pattern is possibly due to the greenhouse effect from enhanced emission of black carbon in this region (by 61%) from 1991-2001. Evidences of incessant increase in temperature during 1990s have also been observed (through chronology of Himalayan Pine) from the contemporaneous surge in tree growth rate (Singh and Yadav 2000). In fact, 50% of the years since 1970 have experienced considerably high solar irradiance and warm phases of ENSO, which is possibly one of the reasons for the considerable rise in temperature throughout the Himalaya (Shekhar et al., 2017) have been added as suggested by the Reviewer in comment 2.

Page 17; lines 480-482 stating Maximum mean annual precipitation is noted during 2015 (615 mm) and minimum during 1946 (244 mm). However, the mean annual precipitation followed a similar trend till 1946 with an increasing thereafter (Fig. 3a;b;c)

have been edited as suggested by the Reviewer in comments 9,10 and 11.

Please also note the supplement to this comment:
https://www.earth-syst-sci-data-discuss.net/essd-2019-122/essd-2019-122-AC1-supplement.pdf
* * *
[Figure]

[Figure]

**Fig. 1.** Annual and seasonal variability in the climate data for the period 1901-2017.

[Figure]

**Fig. 2.** Mean annual temperature and precipitation patterns of CRU-TS derived gridded data
and IMD recorded station at different locations.

(a) Kargil (IMD)

(b) Leh (IMD)

(c) Kargil (CRU-TS)

(d) Leh (CRU-TS)

Precipitation    Temperature

**Fig. 3.** Mean annual temperature and precipitation patterns of IMD recorded station data at Kargil and Leh and their respective CRU-TS derived gridded data.

[Figure]

[Figure]

**Fig. 4.** Percent area loss of the glaciers in the SSB during the period 1971-2017.

[Figure]

**Fig. 5.** Percent length change of the glaciers in the SSB during the period 1971-2017.

---

## Author Comment (AC2) · 18 Dec 2019

Dear Referee, Authors thank the anonymous reviewer for his/her valuable comments and suggestions on our manuscript and the editorial team of the Earth System Science Data for timely processing of the article. Responses to the referee's comments are as follows: Regards. Aparna Shukla.

General comments:

Comment GC1: The study by Shukla et al. entitled, "Temporal inventory of glaciers in the Suru sub-basin, western Himalaya ....." provides very useful data sets of glaciers in

the Suru Sub-Basin in Western Himalaya that are very useful for better understanding the status and fate of the glaciers in the Western Himalaya. The data and manuscript quality is good, except that it would require a major revision to make it in the framework of data paper. Currently, larger focus is on the scientific implications of the data, which is not focus of the journal. While authors have also followed standard methods to process and analyse the data, the methods are not unique.

Response GC1: We agree with your opinion regarding focus of the journal, which aims at publishing articles with original research dataset having the potential to contribute significantly towards the field of Earth Science. In line with the intent of the journal: 1. We have prepared a multi-temporal inventory for four different time periods, which in itself is unique and scarce in the Himalayan region. Apart from addressing the discrepancies, this research also aims to update the data presented in existing inventories (of Suru sub-basin) in order to have a recent and more accurate estimate of glaciers. 2. Inherent data characteristics (glacier area, length, debris cover and snow line altitude changes) have also been assessed to understand the spatial and temporal variability of the glaciers in response to the climate change. 3. Besides, the response of glaciers in Suru sub-basin has also been assessed with respect to other basins of the Himalaya to develop a regional picture. 4. The influence of factors other than climate such as glacier size, regional hypsometry, elevation range, slope, aspect and presence of proglacial lakes have also been evaluated to understand the heterogenous response of the glaciers. To accomplish our objectives, a hybrid methodology is adopted, in which the snow-ice boundaries are mapped using a semi-automatic technique of NDSI and debris coverage through manual digitization. Similar methods of glacier mapping have been employed in other glaciological studies (Bolch et al, 2010; Bhambri et al., 2011; Frey et al., 2012; Chand and Sharma, 2015; Mir et al., 2017; Murtaza and Romshoo, 2015; Molg et al., 2018). In addition, methods have also been employed for estimation of uncertainties which might have introduced from various sources (Hall et al., 2003; Granshaw and Fountain, 2006; Paul et al., 2013;17).

Comment GC2: Overall, large amount of digitization work has been done for this study. However, the Suru basin is a small sub-basin of the Indus river basin, with only 11% of its area is covered with glaciers. So the authors need to substantially revise the manuscript to be useful as a regional representative of Western Himalayan glaciers. Considering the unique scope of the journal, it would therefore, require that the authors to incorporate similar dataset from other distinct basins of Upper Indus Basin to make it more regionally relevant.

Response GC2: Thanks for the suggestion. Suru is actually a sub-basin of Jhelum river basin, which comprises an overall basin area of 50,844 km2 and glacierization of mere 1.4% (733 glaciers) (Bajracharya et al., 2019). In this respect, the Suru sub-basin covers ~9% basin area and 34% glacier count of the entire Jhelum river basin. The prime reason for selection of this very sub-basin for our study purpose was its significant amount of glacier coverage with respect to the entire basin size of the Jhelum. Despite, low percentage coverage (11%), glaciers in the Suru sub-basin show large scale variability locally as well as regionally. Also, the study is unique in itself, as it presents a long time series data of glacier changes and climate patterns, which helps in developing a comprehensive understanding of glacier response on the basin scale (i.e. Suru sub basin). Moreover, existing inventories of the Suru sub-basin as mentioned in the manuscript (Page: 4; lines:132-136) have disparate estimates which need updation. Besides, the Suru sub-basin covers part of two major ranges, i.e., the Greater Himalayan (GHR) and the Ladakh (LR) range, which helps in understanding the existing intra-regional heterogeneity in glacier response and compare it with other basins as well.

The datasets in the manuscript have been processed using a hybrid methodology: Normalized Differential Snow Index (NDSI) for delineation of ice and snow covered boundaries and manual editing for debris cover (Page 10; lines: 231-232 of the original manuscript). The debris cover boundary is manually delineated as no apt technique has been developed till date, which could extract it automatically using optical satellite

images. Moreover, we have also taken assistance of thermal and slope maps for manual digitization of the debris cover boundaries. Similar mapping methodology has been followed by several researchers (Bolch et al., 2010; Chand and Sharma, 2015; Mir et al., 2017; Molg et al., 2018).

Specific comments:

Comment 1:Unlike the Karakoram, the Ladakh Range is not a well known nomenclature. Chudley et al., (2017) have used the Karakoram and Ladakh range, not differentiated about Karakoram and GHR. Mir et al. (2018) have represented it as a part of the GHR. It is therefore, important to define/clarify the same.

Response 1:We agree that the Ladakh range was not a well known nomenclature in the field of glaciology , however, is well recognized in studies pertinent to Himalayan geology (Raz and Honeggar, 1989; Weinberg and Dunlap, 2000; Kirstein et al., 2006; St-Onge et al., 2010; Borneman et al., 2015). Nevertheless, such studies have now become prevalent in glaciology as well, with increase in the number of studies in this region (Schmidt and Nusser, 2012; 2017; Chudley et al., 2017). Chudley et al., (2017) have considered the central and eastern Ladakh range as their research area and have shown that the response of glaciers in these regions is consistent with that in the western Himalaya (to the south), however in contrast to the Karakoram (to the north) Himalaya (Figure R1). In this scenario, our study area covers part of southern Ladakh range (33°54′ to 34°21′ N and 76°00′ to 76°36′ E) and part of Greater Himalayan range (33°43′ to 34°19′ N and 76°37′ to 76°18′ E), lying at the northernmost end of Zanskar range.

Figure R1: Studies conducted in different parts of the western Himalaya (modified after Schmidt and Nusser, 2017)

Mir and Mazeed, (2016), on the other hand, have conducted their study on the Parkachik glacierlocated in the Suru sub-basin. Similar to our study, they have also included the Parkachik/ Kangriz glacier in the GHR (Figure 1 of the original manuscript).

Comment 2: The accuracy of CRU-TS data is not analysed independently. It is critical as the Fig. 3 data looks bit unrealistic. The temperature data indicate dramatic changes after 1990, which needs to be confirmed. Since India Met Department has long term station data in this region as well as gridded data (http://www.imdpune.gov.in/Clim_Pred_LRF_New/Grided_Data_Download.html), it is critical to check the data consistency and conduct error statistics.

Response 2: Thanks for pointing out. In Fig.3 of the original manuscript, the monthly mean precipitation values during the period 1901-2017 had been overestimated due to computational error. This error was introduced due to the variance in formats available for the CRU-TS derived precipitation data and hence was mistaken with the other format (mm/day). The error has now been rectified in the revised manuscript (Page 8; Figure 3d, 3e & 3f). The revised figures (3d, 3e, 3f) show monthly mean precipitation (Jan-Dec) variations of 33 $\pm$14 mm/month in the entire Suru sub-basin, while 37 $\pm$15 mm/month and 30 $\pm$12 mm/month in the GHR and LR, respectively during the period 1901-2017. Figure 3: Annual and seasonal variability in the climate data for the period 1901-2017. (a), (b) and (c) 5 year moving average of the mean annual precipitation (mm) and temperature ($^\circ$C) recorded for 5 grids covering the glaciers in the entire SSB, GHR and LR (sub-regions), respectively during the period 1901-2017. The light and dark grey colored lines depict the respective trend lines for precipitation and temperature conditions during the period 1901-2017. (d), (e) and (f) Monthly mean precipitation and temperature data for the entire SSB, GHR and LR (sub-regions), respectively for the time period 1901-2017.

As rightly indicated by the reviewer, a drastic increase in the mean annual temperature is noticed post 1990, especially from 1995/96 till 2005/06. The mean annual temperature as depicted in figure 3(a), 3(b) and 3(c) shows an overall increase of 0.69$^\circ$C, 0.66$^\circ$C, 0.71$^\circ$C in the Suru sub-basin, GHR and LR, respectively, during period 1990-2017. Infact, the globally averaged combined land and ocean surface temperature data of 1983-2012 period is considered as the warmest 30-year period in the last 1400 years

(IPCC, 2013). This unprecedented rate of warming has been primarily attributed to the rapid scale of industrialization, increase in regional population and anthropogenic activities prevalent during this time period (Bajracharya et al., 2008; IPCC, 2013). Thus, one of the probable reason for this sudden increment in temperature pattern is possibly due to the greenhouse effect from enhanced emission of black carbon in this region (by 61%) from 1991-2001. Evidences of incessant increase in temperature during 1990s have also been observed (through chronology of Himalayan Pine) from the contemporaneous surge in tree growth rate (Singh and Yadav 2000). In fact, 50% of the years since 1970 have experienced considerably high solar irradiance and warm phases of ENSO, which is possibly one of the reasons for the considerable rise in temperature throughout the Himalaya (Shekhar et al., 2017). In order to check data consistency, we have taken up instrument data from nearest stations of Kargil and Leh (due to the unavailability of meteorological stations in the Suru sub-basin) and compared with the CRU-TS derived data for the entire Suru sub-basin during 1901-2002 period (Figure R2).

Figure R2: Mean annual temperature and precipitation patterns of CRU-TS derived gridded data in (a) Suru sub-basin and IMD recorded station at (b) Kargil and (c) Leh. The mean annual temperature pattern of Suru sub-basin shows a near negative trend till 1937, with an increase thereafter. Similar trends have been observed for Kargil and Leh, despite their distant location from the Suru sub-basin (areal distance of Kargil and Leh is ∼63 and 126 km, respectively from the centre of Suru sub-basin). However, it is noteworthy to mention that all the locations had attained maximum mean annual temperature in 1999 (Suru: 2.02°C; Kargil: 6.84°C; Leh: -0.5°C). The results are interesting and we observe an almost similar trend in all the cases (Figure R2),with an accelerated warming post 1995/96. However, the magnitude varies, with longterm mean annual temperature of 0.9, 5.5 and -2.04°C observed in Suru sub-basin, Kargil and Leh, respectively (Figure R2). The possible reason for this difference in their magnitudes could possibly be attributed to their distinct geographical locations and difference in their nature, with former being point, while latter being the interpolated gridded data.

This analysis aptly brings out the bias in the CRU TS gridded data. Majorly the comparison shows that though the gridded data correctly brings out the temporal trends in meteorological data but differ with station data in magnitude (being on lower than the station estimates). This helps us better appreciate the climate variations in Suru subbasin as well since we learn that the reported temperature and precipitation changes are probably on the lower side of the actual variations. Also, we have used the station data, obtained from nearest available IMD sites, i.e., Kargil and Leh and compared with their respective CRU-TS data (mean annual temperature and precipitation).

Figure R3: Mean annual temperature and precipitation patterns of IMD recorded station data at Kargil and Leh and their respective CRU-TS derived gridded data. Though varying in magnitude, the climate data obtained from IMD as well as CRU-TS suggest almost similar trends of temperature and precipitation during the period 1901-2002 for both Kargil and Leh (Figure R3). The annual mean temperature/ precipitation have amounted to 5.5°C/589 mm (IMD) and 2.4°C/315 mm (CRU-TS) in Kargil, while -2.04/279 mm (IMD) and -0.09/ 216 mm (CRU-TS) in Leh during the period 1901-2002 (Figure R3).We observed that climatic variables show lower magnitude in case of CRU-TS as compared to the station data from IMD (except CRU-TS derived temperature data recorded for Leh). The possible reason for this difference between CRU-TS and station data can primarily be attributed to the difference in their nature, with former being point, while latter being a gridded data (0.5° latitude and longitude grid cells). This analysis aptly brings out the bias in the CRU TS gridded data. Majorly the comparison shows that though the gridded data correctly bring out the temporal trends in meteorological data but differ with station data in magnitude (being on lower side than the station estimates). This helps us better appreciate the climate variations in the Suru sub-basin as well, since we learn that the reported temperature and precipitation changes are probably on the lower side of the actual variations.

Comment 3: Considering the large uncertainty involved in Landsat MSS data, it is important to mention the inherent uncertainties while interpreting the temporal variability.

Table 1: include the Scene ID for clarity.

Response 3: We agree with the reviewer. Despite large uncertainties involved in Landsat MSS dataset, we have utilized it to compensate for the data gap in the Corona imageries (covering 40% of the GHR and 58% of the LR glaciers). Previous studies have frequently utilized the Landsat MSS imagery for glacier mapping and analysis for the 1970s period (Pandey and Venkatraman, 2013; Rai et al., 2013; Shangguan et al., 2014; Thakuri et al., 2014; Brahmbhatt et al., 2015; Shukla and Qadir, 2016; Mir et al., 2017). Moreover, we have also accounted for uncertainties using prevalent methods (area and length change uncertainty by Hall et al., 2003 and mapping uncertainty using buffer method by Granshaw and Fountain, 2006)) associated with glacier changes (area and length) using Landsat MSS data and also incorporated the same in the original manuscript (Table 2).

In addition to this, we have now taken 2 glaciers, GL-157 (small, 5.5 km2) and Kangriz glacier (largest, 53 km2) and digitized their boundaries using both the Corona and Landsat MSS imageries. On comparing the glacier boundaries using the two datasets, we noticed that higher uncertainty is associated with the GL-157 (22%) as compared to the Kangriz glacier (0.1%).Considering this, we could say that, though larger in magnitude the uncertainty estimates using Landsat would not affect GHR glaciers much (comparatively larger in size) as compared to the LR (smaller in size) glaciers.

As suggested, the Scene IDs have now been incorporated with theTable1.

Table 1: Detailed specifications of the satellite data utilized in the present study. GB= glacier boundaries, DC=debris cover

Comment 4: Lines 236-240: The procedures used for determining the glacier boundaries are apparently manual digitization. While this is reasonable to undertake manual processing in such complicated areas, it also necessitates a study of uncertainty estimations in such manual work. Authors may also undertake repeatability tests with different analysts to determine repeatability.
Response 4:We have followed a 'hybrid approach', involving normalized difference snow index (NDSI) for delineation of snow-ice boundaries and manual digitization for mapping the debris cover (Page: 10; lines: 231-232 of the original manuscript). Similar mapping methodology has been followed by several researchers (Chand and Sharma, 2015; Mir et al., 2017; Molg et al., 2018). As aptly pointed out by the reviewer, we also agree that manual processing of the database necessitates uncertainty estimation. However, the essence of this work lies in the mapping of the glaciers for multiple (four) time periods by a single analyst, which minimizes the errors to a great extent. While, the repeatability tests are more relevant for studies concerning global scale inventory such as Randolph glacier inventory (RGI), Global land ice measurements from space (GLIMS) and recently Chinese glacier inventory (CGI), where multiple analysts are involved. Nevertheless, we have performed the repeatability tests on the Pensilungpa glacier by delineating its boundary for the year 2017 by 4 different analysts. The test result shows variation in glacier size by all four analysts (17.003 km2, 16.22 km2, 16.59 km2 and 14.67 km2). These values have varied significantly and slightly overestimated from the size estimated using the semi-automatic approach (15.57 km2). The fluctuations in glacier size have varied within the range of 5-10%, i.e., by 9, 4, 6.5 and 6%, respectively), which is acceptable for glacier mapping (Paul et al., 2013).

Comment 5:Lines 272 – 300: The uncertainty assessment is biased with the very limited field validation on only one glacier for a very limited time frame. One issue that needs to be addressed is the reliability of ground truth data when different types of data were used through the nearly 50 years' time period.

Response 5: We agree that very limited field validation has been incorporated for a limited time frame, however, ground based monitoring of the glaciers is difficult and often constrained by extreme conditions prevailing in the Himalayan glaciated terrain. This is very well discerned from the limited field studies (11 in western, 4 in central, 1 in eastern) being conducted in the Himalayan region till date (Pratap et al., 2015; Raina and Srivastava, 2008). In this study, the aim of comparing our results with field

data (initially for 2017) was basically validating the mapping method as data related errors are being already accounted for in the other methods of uncertainty estimation. However, to enhance the reliability of ground data, we have now incorporated field data of the Kangriz glacier as well for year 2018 (obtained from DGPS). On comparing the snout position of the Kangriz glacier derived from DGPS and OLI image, an accuracy of $\pm1.4$ m is obtained. Also, the frontal retreat estimated using DGPS and OLI image is found to be 38.63 $\pm47.8$ and 39.98 $\pm56.6$ m, respectively during the period 2017-18. While the mapping uncertainty of the Kangriz glacier is found to be 0.96%, which shows that our remotely derived estimates matches well with that of field and hence, supports our mapping method. This result has now been incorporated in the revised manuscript (Page: 11; lines: 288-290).

Comment 6: Please discuss why the projective transformation was required for the satellite data sets other than Corona?

Response 6: We have used projective transformation for co-registration of all the images, i.e., Landsat as well as Corona (Page: 10; lines: 227-231 of the original manuscript) in order to maintain uniformity in data processing method. Projective transformation is a novel technique of image registration which projects the 2-dimensional image on the radius and angular coordinates, respectively. Moreover, this method has been used because in contrast to the other methods of image registration, i.e., polynomial and rubber sheeting, projective transformation involves the input reference of DEM which allows the analyst to capture the dynamics of the image and enhances the quality of the two-dimensional data.

Comment 7: Line 328- 330: Categorization of glaciers - is there a scientific standard for categorizing the glaciers in the different categories or was more based on the author's selectivity? Check DeBeer and Sharp (2009, Journal of Glaciology). Since the data descriptions needs to be internationally consistent, may revise.

Response 7: It is a welcome suggestion. However, glacier size is a variable parameter which fluctuates from basin to basin and hence, cannot be standardized globally or for a particular region. Moreover, to the best of our knowledge, there is no scientific standard for categorizing the glaciers and for this study, it is entirely based on investigators selectivity. DeBeer and Sharp, (2009) have categorized small glaciers in the British Columbia as per the size distribution of the glacier in the region, i.e., <0.4 km2 as very small and 0.4-5 km2 as large glaciers. However, in the Himalayan region different studies have used different size class for the glaciers (Table RT1). Owing to this heterogeneity in glacier size classification, we have not followed any particular study, but, have given a separate categorization (Page:12,lines:328-330 of the original manuscript).

Table RT1: Size distribution of glaciers in different basins of the Himalaya.

Comment 8: Statistical significance could be included to explain the effect of spatial characteristics (size, aspect, debris cover) or any difference spatial control over LR and GHR.

Response 8: Thanks for the suggestion. We understand the reviewer's point that GHR and LR comprises of different glaciers having distinct morphology. However, in our analysis, we have taken into account the change in glacier parameters in terms of percentage, which is normalized. Hence, the data is not susceptible to any biases. Moreover, we have followed a sequential method of data analysis: in which all the glaciers are first investigated for parametric changes and we observe regional heterogeneity in glacier response. Thereafter, we went for understanding the possible controls on the reported changes, in which we noted that the glacier response is primarily influenced by climate variability (statistical significance taken into account). The study also confirms the possible controls of non-climatic factors (in terms of percentage) on heterogeneous glacier response. However, we have now incorporated the statistical significance to explain the effect of spatial characteristics (size, slope, debris cover and elevation) over LR and GHR (Supplementary material of the revised manuscript). For this, the non-climatic factors were subsequently correlated with the change in glacier dimensional

parameters, i.e., area change and retreat using some statistical tests (Figure R4a,b; Table RT2). In the statistical analysis, the variables were initially tested for normality and visual inspection of the histogram. The test showed normal distribution for nearly all the variables and the correlations were found to be significant at $\alpha < 0.05$ (except for mean elevation). These correlations also showed the presence of few outliers (not removed in this study), which indicate the possible role of any other factor due to which these glaciers have deviated from the general trend of area loss and retreat (Figure R4a;b).

Table RT2:Correlation (r) and Pearson's correlation (p) coefficient computed between non-climatic factors (size, slope, debris cover and elevation) and glacier changes (% deglaciation & retreat rate). These relationships were found to be significant at $\alpha < 0.05$ (Except for mean elevation: Italicized).

Figure R4a. Scatter plots displaying the relation between topographic factors with percent deglaciation during the period 1971-2017. All the relationships were found to be significant at confidence level, i.e., $\alpha<0.05$ (Except mean elevation).

Figure R4b. Scatter plots displaying the relation between topographic factors with retreat rate during the period 1971-2017. All the relationships were found to be significant at confidence level, i.e., $\alpha<0.05$ (Except mean elevation).

REFERENCES Bajracharya, S. R., Mool, P. K., Shrestha, B. R.: Global climate change and melting of Himalayan glaciers. Melting glaciers and rising sea levels: Impacts and implications, Prabha Shastri Ranade (ed), The Icfai's University Press, India, 28–46, 2008.

Basnett, S., Kulkarni, A.V. and Bolch, T.: The influence of debris cover and glacial lakes on the recession of glaciers in Sikkim Himalaya, India, Journal of Glaciology, 59, 1035-1046, https://doi.org/10.3189/2013JoG12J184, 2013.

Bhambri, R., Bolch, T., Chaujar, R. K., and Kulshreshtha, S. C.: Glacier changes in

the Garhwal Himalaya, India, from 1968 to 2006 based on remote sensing, Journal of Glaciology, 57, 543–556, https://doi.org/10.3189/002214311796905604, 2011.

Bolch, T., Yao, T., Kang, S., Buchroithner, M. F., Scherer, D., Maussion, F., Huintjes, E., and Schneider, C.: A glacier inventory for the western Nyainqentanglha Range and the Nam Co Basin, Tibet, and glacier changes 1976–2009, The Cryosphere, 4, 419-433, https://doi.org/10.5194/tc-4-419-2010, 2010.

Borneman, N. L., Hodges, K.V., Van Soest, M. C., Bohon, W., Wartho, J. A., Cronk, S. S. and Ahmad, T.: Age and structure of the Shyok suture in the Ladakh region of northwestern India: implications for slip on the Karakoram fault system, Tectonics, 34, 2011-2033, 2015.

[revised manuscript text omitted]

Schmidt, S. and Nüsser, M.: Changes of high altitude glaciers from 1969 to 2010 in the Trans-Himalayan Kang Yatze Massif, Ladakh, northwest India. Arct. Antarct. Alp. Res., 44, 107–121, 2012. Schmidt, S. and Nusser, M.: Changes of High Altitude Glaciers in the Trans-Himalaya of Ladakh over the Past Five Decades (1969–2016), Geosciences,

7, 27, https://doi.org/10.3390/geosciences7020027, 2017.

Shangguan, D., Liu, S., Ding., Y., Wu, L., Deng, W., Guo, W., Wang, Y., Xu, J., Yao, X., Guo, Z. and Zhu, W.: Glacier changes in the Koshi River basin, central Himalaya, from 1976 to 2009,derived from remote-sensing imagery, Annals of glaciology, 55, 61-68, https://doi.org/10.3189/2014AoG66A057, 2014.

Shekhar, M., Bhardwaj, A., Singh, S., Ranhotra1, P. S., Bhattacharyya, A., Pal, A. K., Roy, I., Martín-Torres, F. J. and Zorzano, M.P.: Himalayan glaciers experienced significant mass loss during later phases of little ice age, Scientific Reports, 7, 1-14, 2017.

Shukla, A. and Qadir, J.: Differential response of glaciers with varying debris cover extent: evidence from changing glacier parameters, International Journal of Remote Sensing, 37, 2453–2479, http://doi.org/10.1080/01431161.2016.1176272, 2016.

Singh, J. and Yadav, R. R.: Tree-ring indications of recent glacier fluctuations in Gangotri, western Himalaya, India, Current Science, 79(11), 1598–1601, 2000.

St-Onge, M. R., Rayner, N. and Searle, M. P.: Zircon age determinations for the Ladakh batholith at Chumathang (Northwest India): implications for the age of the India–Asia collision in the Ladakh Himalaya, Tectonophysics, 495, 171-183, 2010.

Thakuri, S., Salerno, F., Smiraglia, C., Bolch, T., Agata, C. D., Viviano, G. and Tartari, G.: Tracing glacier changes since the 1960s on the south slope of Mt. Everest (central Southern Himalaya) using optical satellite imagery, The Cryosphere, 8, 1297-1315, http://doi.org/10.5194/tc-8-1297-2014, 2014.

Weinberg, R. F. and Dunlap, W. J.: Growth and deformation of the Ladakh Batholith, Northwest Himalayas: implications for timing of continental collision and origin of calc-alkaline batholiths, The Journal of Geology, 108, 303-320, 2000.

Please also note the supplement to this comment:

https://www.earth-syst-sci-data-discuss.net/essd-2019-122/essd-2019-122-AC2-supplement.pdf

**Study area:**

(After Schmidt and Nusser, 2017)

☐ Schmidt and Nusser, 2017    ☐ Chudley et al., 2017

☐ Shukla et al., 2019 (Present study)    ⬭ Mir et al., 2016

**Fig. 1.** Figure R1: Studies conducted in different parts of the western Himalaya (modified after Schmidt and Nusser, 2017)

[Figure]

**Fig. 2.** Figure 3: Annual and seasonal variability in the climate data for the period 1901-2017.

[Figure]

**Fig. 3.** Mean annual temperature and precipitation patterns of CRU-TS derived gridded data in (a) Suru sub-basin and IMD recorded station at (b) Kargil and (c) Leh.

**Fig. 4.** Mean annual temperature and precipitation patterns of IMD recorded station data at Kargil and Leh and their respective CRU-TS derived gridded data.

**Fig. 5.** Scatter plots displaying the relation between topographic factors with percent deglaciation during the period 1971-2017.

[Figure]

**Fig. 6.** . Scatter plots displaying the relation between topographic factors with retreat rate during the period 1971-2017.

---

## Author Response (AR1)

Dear Editor,

Please find enclosed the revised manuscript (Manuscript ID: essd-2019-122) entitled "Temporal inventory of glaciers in the Suru sub-basin, western Himalaya: Impacts of the regional climate variability". On behalf of all the authors, I would like to thank the editorial team of the Earth System Science Data for timely processing of the manuscript and two anonymous referees for their critical reviews and the following constructive suggestions for improving the original manuscript:

- Enhance the climate study by incorporating error statistics, comparative analysis of used gridded data with in-situ data and explanation for the obtained trends.
- Revise the figures so that the readers could better relate the text with the respective figures and avoid confusion.
- Incorporate statistical significance of non-climatic parameters (size, debris cover, elevation, slope) to explain the effect of spatial characteristics on LR and GHR glaciers.
- Add more data from the field to ensure data consistency.

In line with the listed major suggestions, we have addressed to all the comments and now revised the manuscript accordingly:

- Climate study is enhanced by adding meteorological data (temperature and precipitation) derived from nearby stations of Kargil and Leh (IMD and the changes have been incorporated in the revised manuscript.
- Figures: 3, 6 and 7 (revised manuscript) have been simplified and updated as suggested by the reviewers.
- Statistical tests of the non climatic factors (size, slope, mean elevation, debris cover) have now been performed and incorporated in the revised manuscript.
- Field data obtained from other glacier has been incorporated in the revised manuscript.

Our responses to the Reviewers comments and revised manuscript are attached with this letter. I confirm that all the authors have approved the submission of this manuscript and it is not currently under consideration for publication elsewhere.

Thanks for your consideration.

Yours Sincerely,

Aparna Shukla

**Response to the comments on "Temporal inventory of glaciers in the Suru sub-basin, western Himalaya: Impacts of the regional climate variability" by Shukla et al., 2019**

*Referee # 1:*

**Comment 1:** Long-term climate data presentation and analysis needs attention. Page 8, Line 183; Mean precipitation of the SSB for the period 1901-2017 has been 393 ±*76* mm. However, if we see plots in figure 3(d), 3(e) and 3(f), monthly mean precipitation for the same period are quite high indicating high precipitation during the same period.

**Response 1:** Thanks for pointing out. The annual average precipitation for the Suru sub-basin amounts to 393 ±*76* mm during the period 1901-2017. However, the monthly mean precipitation values during the same period had been overestimated due to computational error. This error was introduced due to the variance in formats available for the CRU-TS derived precipitation data and hence was mistaken with the other format (mm/day). The error has now been rectified in the revised manuscript (Page 8; Figure 3d, 3e & 3f). The revised figures (3d, 3e, 3f) show monthly mean precipitation (Jan-Dec) variations of 33 ±*14* mm/month in the entire Suru sub-basin, while 37 ±*15* mm/month and 30 ±*12* mm/month in the GHR and LR, respectively during the period 1901-2017.

[Figure]

Figure 3 (revised manuscript): Annual and seasonal variability in the climate data for the period 1901-2017. (a), (b) and (c) 5 year moving average of the mean annual precipitation (mm) and temperature (°C) recorded for 5 grids covering the glaciers in the entire Suru sub-basin (SSB), Greater Himalayan Range (GHR) and Ladakh Range (LR) (sub-regions), respectively during the period 1901-2017. The light and dark grey colored lines depict the respective trend lines for precipitation and temperature conditions during the period 1901-2017. (d), (e) and (f) Monthly mean precipitation and temperature data for the entire SSB, GHR and LR (sub-regions), respectively for the time period 1901-2017.

**Comment 2:** Figure 3(a), 3(b) and 3(c) shows continuous increase in the temperature during the period 1995-96 onwards till 2005-06. It shows sudden change in the temperature pattern. The reason for the sudden shift in temperature pattern should be discussed. It will be interesting to see the temperature pattern of the IMD recorded data at Leh or any other in-situ recorded data in the study region during the same period.

**Response 2:** Agreed. The mean annual temperature depicted in figure 3(a), 3(b) and 3(c) shows an overall increase of 0.71°C, 0.72°C, 0.71°C in the Suru sub-basin, GHR and LR, respectively, during 1995/96 till 2005/06 period as mentioned by the referee. The globally averaged combined land and ocean surface temperature data of 1983-2012 period is considered as the warmest 30-year period in the last 1400 years (IPCC, 2013). This unprecedented rate of warming has been primarily attributed to the rapid scale of industrialization, increase in regional population and anthropogenic activities prevalent during this time period (Bajracharya et al., 2008; IPCC, 2013). Thus, one of the probable reason for this sudden increment in temperature pattern is possibly due to the greenhouse effect from enhanced emission of black carbon in this region (by 61%) from 1991-2001 (Sahu et al., 2008). Evidences of incessant increase in temperature during 1990s have also been observed (through chronology of Himalayan Pine) from the contemporaneous surge in tree growth rate (Singh and Yadav 2000). In fact, 50% of the years since 1970 have experienced considerably high solar irradiance and warm phases of ENSO, which is possibly one of the reasons for the considerable rise in temperature throughout the Himalaya (Shekhar et al., 2017). The same has now been discussed in the revised manuscript as suggested (Page: 19, lines: 502-512).

Due to the unavailability of in-situ climate dataset for the Suru sub-basin, station data is obtained from nearest stations of Kargil and Leh and compared with the CRU-TS derived data for the entire Suru sub-basin during 1901-2002 period.

[Figure]

Figure R1(Figure 4 of the revised manuscript): Mean annual temperature and precipitation patterns of CRU-TS derived gridded data and IMD recorded station at different locations.

The mean annual temperature pattern of Suru sub-basin shows a near decreasing trend till 1936, with an increase thereafter. Similar trends have been observed for Kargil and Leh, despite their distant location from the Suru sub-basin (areal distance of Kargil and Leh is ~63 and 126 km, respectively from the centre of Suru sub-basin). However, it is noteworthy to mention that all the locations had attained maximum mean annual temperature in 1999 (Suru: 2.02°C; Kargil: 6.84°C; Leh: -0.5°C).

Indeed, these results are interesting and we observe an almost similar trend in all the cases (Figure R1), with an accelerated warming post 1995/96. However, the magnitude varies, with longterm mean annual temperature of 0.9, 5.5 and -2.04°C observed in Suru sub-basin, Kargil and Leh, respectively (Figure R1). While the change (increase) in mean annual temperature observed during the same period, i.e., 1901-2002 is found to be 0.34, 0.13 and 0.44 °C in Suru sub-basin, Kargil and Leh, respectively. The possible reason for this difference in their magnitudes could possibly be attributed to their distinct geographical locations and difference in their nature, with former being point, while latter being the interpolated gridded data.

**Comment 3:** A comparison of the CRU data with in-situ (temperature and precipitation) in the study region will provide information about the biases in the CRU data.

**Response 3:** Agreed. Due to the unavailability of meteorological observatories in the Suru sub-basin, station data is obtained from nearest available IMD sites, i.e., Kargil and Leh and compared with their respective CRU-TS data (mean annual temperature and precipitation).

[Figure]

Figure R2 (Figure 5 of the revised manuscript): Analysis of meteorological (mean annual temperature and precipitation) datasets derived from Indian Meteorological Department (IMD) stations at (a) Kargil & (b) Leh and the respective [(c) Kargil and (d) Leh] gridded data obtained from climate research unit (CRU)-time series (TS).

Though varying in magnitude, the climate data obtained from IMD as well as CRU-TS suggest almost similar trends of temperature and precipitation during the period 1901-2002 for both Kargil and Leh (Figure R2). The annual mean temperature/ precipitation have amounted to 5.5°C/589 mm (IMD) and 2.4°C/315 mm (CRU-TS) in Kargil, while -2.04/279 mm (IMD) and -0.09/ 216 mm (CRU-TS) in Leh during the period 1901-2002 (Figure R2).We observed that climatic variables show lower magnitude in case of CRU-TS as compared to the station data from IMD (except CRU-TS derived temperature data recorded for Leh).The possible reason for this difference between CRU-TS and station data can primarily be attributed to the difference in their nature, with former being point, while latter being a gridded data (0.5° latitude and longitude grid cells). This analysis aptly brings out the bias in the CRU TS gridded data. Majorly the comparison shows that though the gridded data correctly bring out the temporal trends in meteorological data but differ with station data in magnitude (being on lower side than the station estimates). This helps us better appreciate the climate variations in the Suru sub-basin as well, since we learn that the reported temperature and precipitation changes are probably on the lower side of the actual variations.

This analysis has been incorporated in the climate analysis section of the revised manuscript (Page: 11-13; lines: 276-308).

**Comment 4:** Page 13, Line 339; How authors will explain the mean slope variation of 16.2° ±*71*° to 41° ±*66*°?

**Response 4:** Thanks for pointing it. In this study, range of slope was reported initially depicting minimum and maximum variations in the overall data, i.e., in 16.2° ±*71*°, 16.2° was the average minimum slope and 71° was the deviation in this minimum slope considering the entire basin. Similarly, in 41° ±*66*°, 41° was the average maximum slope while 66° was the deviation in this maximum slope considering the entire basin. However, we now realize this form of data representation misleading. Therefore, the mean slope of the GHR and LR glaciers have now been mentioned, which has varied from 24 ±*6*° to 25 ±*6*°, respectively. The same has now been incorporated in the revised manuscript (Page: 15, Line: 380).

**Comment 5:** Figure 4(a) Frequency distribution histogram depicting maximum frequency in the percent area change between 0.52-0.97. How it concludes that majority of the glaciers have undergone an area loss of 3.3%.

**Response 5:** The statement mentioning that the majority of the glaciers have undergone area change of 3.3% was based on mid-point of a legend category (0.8-6%) as shown in the chloropleth map. This was misleading as the categories of percent area change depicted in histogram differed from those shown in the chloropleth map. However, now we have simplified the histogram and the chloropleth map by keeping same divisions (range of percent area change) for both.

In the revised Figure 6a, it may be observed that majority of the glaciers have undergone area change of the range 6-12% and same is depicted in the chloropleth.

[Figure]

Figure 6 (revised): (a) Percent area loss of the glaciers in the SSB during the period 1971-2017. Frequency distribution histogram depicting that majority of the glaciers have undergone an area loss in the range 6-12%. (b) Hypsometric distribution of glacier area in the GHR and LR regions during the period (I) 1971-2000 and (II) 2000-2017. (A), (B), (C) and (D) insets in (II) shows the significant change in area at different elevation range of the GHR and LR glaciers.

**Comment 6:** Figure 5; Majority of the glaciers have undergone length change of 5% is not seen in the frequency distribution histogram.

**Response 6:** The statement mentioning that the majority of the glaciers have undergone length change of 5% was based on mid-point of a legend category (0.9-8%) as shown in the chloropleth map. This was misleading as the categories of percent length change depicted in histogram differed from those shown in the chloropleth map. However, now we have simplified the histogram and the chloropleth map by keeping same divisions (range of percent length change) for both.

In the revised Figure 7, it may be observed that majority of the glaciers have undergone length change of the range 6-14% and same is depicted in the chloropleth.

[Figure]

Figure 7 (revised): Percent length change of the glaciers in the SSB during the period 1971-2017. Frequency distribution histogram showing that majority of the glaciers have undergone length change in the range 6-14%.

**Comment 7:** What could be the possible reasons of decrease in SLA in LR glaciers despite of increase in temperature and retreat in glacier length in the region?

**Response 7:** Yes, if we simply try to equate the absolute temperature change in LR with the overall SLA and/or length changes observed in this region then the results might seem counter-intuitive. However, such is not the case. While SLA [often used as a reliable proxy for glacier mass balance changes (Guo et al., 2014)] responds directly to the changes in meteorological variables mostly temperature, length changes or retreat are much delayed response of the glaciers towards climate change (Bolch et al., 2012; Paul et al., 2017). Besides, glacier retreat is often strongly influenced by the local snout characteristics and conditions such as presence of proglacial lakes, supraglacial debris coverage and differential shadowing (Sakai, 2012; Shukla and Qadir, 2016; Garg et al., 2017). For these reasons, SLA and retreat trends may not always be in-sync.

Coming to the reported increase in temperature in the LR, this increase has been estimated using following formulation which takes into account longterm mean and trends of entire temperature data series in the form of Sen's slope.

$$ChangeinTemperatureandPrecipitation = (\beta * L)/M$$

where β is Sen's slope estimator, $L$ is length of period and $M$ is the long term mean.

Contrary to this, the reported SLA changes are simple difference between the average SLAs of 1977 and 2017. Thus, the SLA changes seem counter-intuitive to the temperature variations and do not correlate well with it. However, if we break this long time frame of 40 years (1977–2017) into shorter time periods then we find that the SLA in LR had been responding excellently to the ongoing temperature changes (Table R1). Also, the SLA and temperature changes have, as expected, high negative correlation with each other (i.e., -0.82).

Table RT1: Period wise variations in SLA of the LR glaciers and changes in temperature conditions during the corresponding time interval.

| Time Period | SLA change (m) | Temperature change (°C) |
|---|---|---|
| 1977-94 | 12.55 | Decrease in temp by 0.11 |
| 1994-2000 | -103.31 | Increase in temp by 0.71 |
| 2000-17 | 108.96 | Increase in temp by 0.02 |

Where, (-ve) sign: rise in SLA, (+ve) sign: decrease in SLA

**Comment 8:** Page 16; Line 405; there is a large difference in the number of glaciers reported in the sub basin by earlier researchers and reported in the present paper. It needs discussion and possible reasons. Is there any difference in defining a glacier?

**Response 8:** Statistics of the year 2000 reveal a total of 240 glaciers in the Suru sub-basin (Page 16; Line: 404 of the original manuscript).This is, though comparable with that reported by Sangewar and Shukla, (2009) i.e. 284, varies drastically from SAC report, (2016) and RGI (2 different analysts) [110 and (514 & 304), respectively].One possible reason could be the difference in methodology adopted for glacier delineation leading to systematic errors (Page: 16; Lines: 410-411 of the original manuscript). Secondly, the involvement of multiple analysts may introduce random errors, as in case of RGI (Page: 16; Lines: 411-412 of the original manuscript).

Yes, as already pointed out there is a difference in defining a glacier in these studies, which is yet another plausible reason for introducing the bias in the glacier count. RGI have provided separate glacier id to each polygon in the Sub-basin, which might be the reason for overestimation of glaciers. While no such information regarding the definition of glacier has been provided in the SAC report, (2016). However, in this study, the glacierets / tributary glaciers contributing to the main trunk are considered as a single glacier entity, which is a standard procedure for assigning the glacier id. The statement was somehow missing from the original manuscript which would have created the confusion, therefore, now it has been incorporated in the revised manuscript (Page 10, lines 239-240).

**Comment 9:** Page 18, Line 462; statement 'However a sudden decrease in the precipitation anomaly is observed in the year 2016 with an increase thereafter', it is not clear to me that Figure 3(a), (b) and (c) are showing 'precipitation' or 'precipitation anomaly'? Year 2016 is missing in the Figure.

**Response 9:** Figure 3(a) (b) and (c) are showing *'5 year moving average of average annual precipitation'*. The statement mentioned in the original manuscript regarding 'precipitation anomaly' was previously included in the graphs. However, these graphs were changed (with different mode of representation) later owing to more information shown by present graphs included in the manuscript (Figure 3). These lines should have been removed from the text as well. We regret their inclusion. The vertical bars show 5 year moving average of mean annual precipitation during the period 1901-2017.

**Comment 10:** Page 18, Line 462-463; statement regarding mean annual precipitation is not clear if I look at Figure.

**Response 10:** Similar to Response 9

**Comment 11:** Page 18, Line 463-464; 'temperature and precipitation anomaly' not understood.

**Response 11:** Thanks for pointing out. The statement mentioned in the original manuscript regarding 'temperature and precipitation anomaly' was previously included in the graphs. However, these graphs were changed (with different mode of representation) later owing to more information shown by present graphs included in the manuscript (Figure 3). These lines should have been removed from the text as well. We regret their inclusion.The statement has now been edited to "Besides these general trends in mean annual temperature and precipitation, an overall absolute increase in the mean annual temperature ($T_{max}$ & $T_{min}$) and precipitation data have been noted as 0.77 °C (0.25 °C & 1.3 °C) and 158 mm, respectively during the period 1901-2017" (Revised manuscript; Page 19; lines 512-514).

**Comment 12:** It is advised to draw a trend line for temperature and precipitation variation in Figure 3.

**Response 12:** Thanks for the suggestion. A trend line for temperature and precipitation variations have now been added in Figure 3 of the revised manuscript.

**Comment 13:** Page 18, Line 466; 'Percentage increase in the average, maximum and minimum temperature observed to be 99,12 and 17%', generally temperature variation is not shown in percentage. I will give an example, if mean temperature varies from 0.1°C to 0.2°C for one year and next year it drops to 0.1°C again, should one conclude that temperature variation was 100%

increasing for the first year and 100% decreasing for next year. Statement will be misleading, since the temperature variation was minimal. If the unit of temperature changes from °C to K, then still the statement will hold good? It is advised not to represent temperature variation in % throughout the manuscript.

**Response 13:** Agreed. As suggested, we have now reported the temperature and precipitation changes in absolute form rather than in percentage.

*Referee # 2:*

**General comments:**

**Comment GC1:** The study by Shukla et al. entitled, "Temporal inventory of glaciers in the Suru sub-basin, western Himalaya ....." provides very useful data sets of glaciers in the Suru sub-basin in Western Himalaya that are very useful for better understanding the status and fate of the glaciers in the Western Himalaya. The data and manuscript quality is good, except that it would require a *major revision* to make it in the framework of data paper. Currently, larger focus is on the scientific implications of the data, which is not focus of the journal. While authors have also followed standard methods to process and analyze the data, the methods are not unique.

**Response GC1**: We agree with your opinion regarding focus of the journal, which aims at publishing articles with original research dataset having the potential to contribute significantly towards the field of Earth Science. In line with the intent of the journal:

1. We have prepared a multi-temporal inventory for four different time periods, which in itself is unique and scarce in the Himalayan region. Apart from addressing the discrepancies, this research also aims to update the data presented in existing inventories (of Suru sub-basin) in order to have a recent and more accurate estimate of glaciers.

2. Inherent data characteristics (glacier area, length, debris cover and snow line altitude changes) have also been assessed to understand the spatial and temporal variability of the glaciers in response to the climate change.

3. Besides, the response of glaciers in Suru sub-basin has also been assessed with respect to other basins of the Himalaya to develop a regional picture.

4. The influence of factors other than climate such as glacier size, regional hypsometry, elevation range, slope, aspect and presence of proglacial lakes have also been evaluated to understand the heterogeneous response of the glaciers.

To accomplish our objectives, a hybrid methodology is adopted, in which the snow-ice boundaries are mapped using a semi-automatic technique of NDSI and debris coverage through manual digitization. Similar methods of glacier mapping have been employed in other glaciological studies (Bolch et al, 2010; Bhambri et al., 2011; Frey et al., 2012; Chand and Sharma, 2015; Mir et al., 2017; Murtaza and Romshoo, 2015; Molg et al., 2018). In addition, methods have also been employed for estimation of uncertainties which might have introduced from various sources (Hall et al., 2003; Granshaw and Fountain, 2006; Paul et al., 2013;17).

**Comment GC2:** Overall, large amount of digitization work has been done for this study. However, the Suru basin is a small sub-basin of the Indus river basin, with only 11% of its area is covered with glaciers. So the authors need to substantially revise the manuscript to be useful as a regional representative of Western Himalayan glaciers. Considering the unique scope of the journal, it would therefore, require that the authors to incorporate similar dataset from other distinct basins of Upper Indus Basin to make it more regionally relevant.

**Response GC2:** Thanks for the suggestion. Suru is actually a sub-basin of Jhelum river basin, which comprises an overall basin area of 50,844 km$^2$ and glacierization of mere 1.4% (733 glaciers) (Bajracharya et al., 2019). In this respect, the Suru sub-basin covers ~9% basin area and 34% glacier count of the entire Jhelum river basin. The prime reason for selection of this very sub-basin for our study purpose was its significant amount of glacier coverage with respect to the entire basin size of the Jhelum.

Despite, low percentage coverage (11%), glaciers in the Suru sub-basin show large scale variability locally as well as regionally. Also, the study is unique in itself, as it presents a long time series data of glacier changes and climate patterns, which helps in developing a comprehensive understanding of glacier response on the basin scale (i.e. Suru sub basin). Moreover, existing inventories of the Suru sub-basin as mentioned in the manuscript (Page 4; lines:132-136 of the original manuscript) have disparate estimates which need updation. Besides, the Suru sub-basin covers part of two major ranges, i.e., the Greater Himalayan (GHR) and the

Ladakh (LR) range, which helps in understanding the existing intra-regional heterogeneity in glacier response and compare it with other basins as well.

The datasets in the manuscript have been processed using a hybrid methodology: Normalized Differential Snow Index (NDSI) for delineation of ice and snow covered boundaries and manual editing for debris cover (Page 10; lines: 231-232 of the original manuscript). The debris cover boundary is manually delineated as no apt technique has been developed till date, which could extract it automatically using optical satellite images. Moreover, we have also taken assistance of thermal and slope maps for manual digitization of the debris cover boundaries. Similar mapping methodology has been followed by several researchers (Bolch et al., 2010; Chand and Sharma, 2015; Mir et al., 2017; Molg et al., 2018).

**Specific comments:**

**Comment 1:** Unlike Karakoram, the Ladakh Range is not a well known nomenclature. Chudley et al., (2017) have used the Karakoram and Ladakh range, not differentiated about Karakoram and GHR. Mir et al. (2018) have represented it as a part of the GHR. It is therefore, important to define/clarify the same.

**Response 1:** We agree that the Ladakh range was not a well known nomenclature in the field of glaciology, however, is well recognized in studies pertinent to Himalayan geology (Raz and Honeggar, 1989; Weinberg and Dunlap, 2000; Kirstein et al., 2006; St-Onge et al., 2010; Borneman et al., 2015). Nevertheless, such studies have now become prevalent in glaciology as well, with increase in the number of studies in this region (Schmidt and Nusser, 2012; 2017; Chudley et al., 2017).

Chudley et al., (2017) have considered the central and eastern Ladakh range as their research area and have shown that the response of glaciers in these regions is consistent with that in the western Himalaya (to the south), however in contrast to the Karakoram (to the north) Himalaya (Figure R3). In this scenario, our study area covers part of southern Ladakh range (33°54´ to 34°21´ N and 76°00´ to 76°36´ E) and part of Greater Himalayan range (33°43´ to 34°19´ N and 76°37´ to 76°18´ E), lying at the northernmost end of Zanskar range.

[Figure]

Figure R3: Studies conducted in different parts of the western Himalaya (modified after Schmidt and Nusser, 2017)

Mir and Mazeed, (*2016*), on the other hand, have conducted their study on the Parkachik glacier located in the Suru sub-basin. Similar to our study, they have also included the Parkachik/ Kangriz glacier in the GHR (Figure 1 of the original manuscript).

**Comment 2:** The accuracy of CRU-TS data is not analysed independently. It is critical as the Fig. 3 data looks bit unrealistic. The temperature data indicate dramatic changes after 1990, which needs to be confirmed. Since India Met Department has long term station data in this region as well as gridded data (http://www.imdpune.gov.in/Clim_Pred_LRF_New/Grided_Data_Download.html), it is critical to check the data consistency and conduct error statistics.

**Response 2:** Thanks for pointing out. In Fig.3 of the original manuscript, the monthly mean precipitation values during the period 1901-2017 had been overestimated due to computational error. This error was introduced due to the variance in formats available for the CRU-TS derived precipitation data and hence was mistaken with the other format (mm/day). The error has now been rectified in the revised manuscript (Page 8; Figure 3d, 3e & 3f). The revised figures (3d, 3e,

3f) show monthly mean precipitation (Jan-Dec) variations of 33 ±14 mm/month in the entire Suru sub-basin, while 37 ±15 mm/month and 30 ±12 mm/month in the GHR and LR, respectively during the period 1901-2017.

As rightly indicated by the reviewer, a drastic increase in the mean annual temperature is noticed post 1990, especially from 1995/96 till 2005/06. The mean annual temperature as depicted in figure 3(a), 3(b) and 3(c) shows an overall increase of 0.69°C, 0.66°C, 0.71°C in the Suru sub-basin, GHR and LR, respectively, during period 1990-2017. In fact, the globally averaged combined land and ocean surface temperature data of 1983-2012 period is considered as the warmest 30-year period in the last 1400 years (IPCC, 2013). This unprecedented rate of warming has been primarily attributed to the rapid scale of industrialization, increase in regional population and anthropogenic activities prevalent during this time period (Bajracharya et al., 2008; IPCC, 2013). Thus, one of the probable reason for this sudden increment in temperature pattern is possibly due to the greenhouse effect from enhanced emission of black carbon in this region (by 61%) from 1991-2001. Evidences of incessant increase in temperature during 1990s have also been observed (through chronology of Himalayan Pine) from the contemporaneous surge in tree growth rate (Singh and Yadav 2000). In fact, 50% of the years since 1970 have experienced considerably high solar irradiance and warm phases of ENSO, which is possibly one of the reasons for the considerable rise in temperature throughout the Himalaya (Shekhar et al., 2017).

In order to check data consistency, we have taken up instrument data from nearest stations of Kargil and Leh (due to the unavailability of meteorological stations in the Suru sub-basin) and compared with the CRU-TS derived data for the entire Suru sub-basin during 1901-2002 period (Figure R1).

The mean annual temperature pattern of Suru sub-basin shows a near decreasing trend till 1936, with an increase thereafter. Similar trends have been observed for Kargil and Leh, despite their distant location from the Suru sub-basin (areal distance of Kargil and Leh is ~63 and 126 km, respectively from the centre of Suru sub-basin). However, it is noteworthy to mention that all the locations had attained maximum mean annual temperature in 1999 (Suru: 2.02°C; Kargil: 6.84°C; Leh: -0.5°C).

The results are interesting and we observe an almost similar trend in all the cases (Figure R1),with an accelerated warming post 1995/96. However, the magnitude varies, with longterm mean annual temperature of 0.9, 5.5 and -2.04°C observed in Suru sub-basin, Kargil and Leh, respectively (Figure R1). The possible reason for this difference in their magnitudes could possibly be attributed to their distinct geographical locations and difference in their nature, with former being point, while latter being the interpolated gridded data.

Also, we have used the station data, obtained from nearest available IMD sites, i.e., Kargil and Leh and compared with their respective CRU-TS data (mean annual temperature and precipitation).

Though varying in magnitude, the climate data obtained from IMD as well as CRU-TS suggest almost similar trends of temperature and precipitation during the period 1901-2002 for both Kargil and Leh (Figure R2). The annual mean temperature/ precipitation have amounted to 5.5°C/589 mm (IMD) and 2.4°C/315 mm (CRU-TS) in Kargil, while -2.04/279 mm (IMD) and -0.09/ 216 mm (CRU-TS) in Leh during the period 1901-2002 (Figure R2).We observed that climatic variables show lower magnitude in case of CRU-TS as compared to the station data from IMD (except CRU-TS derived temperature data recorded for Leh). The possible reason for this difference between CRU-TS and station data can primarily be attributed to the difference in their nature, with former being point, while latter being a gridded data (0.5° latitude and longitude grid cells). This analysis aptly brings out the bias in the CRU TS gridded data. Majorly the comparison shows that though the gridded data correctly bring out the temporal trends in meteorological data but differ with station data in magnitude (being on lower side than the station estimates). This helps us better appreciate the climate variations in the Suru sub-basin as well, since we learn that the reported temperature and precipitation changes are probably on the lower side of the actual variations.

**Comment 3:** Considering the large uncertainty involved in Landsat MSS data, it is important to mention the inherent uncertainties while interpreting the temporal variability. Table 1: include the Scene ID for clarity.

**Response 3:** We agree with the reviewer. Despite large uncertainties involved in Landsat MSS dataset, we have utilized it to compensate for the data gap in the Corona imageries (covering 40% of the GHR and 58% of the LR glaciers). Previous studies have frequently utilized the

Landsat MSS imagery for glacier mapping and analysis for the 1970s period (Pandey and Venkatraman, 2013; Rai et al., 2013; Shangguan et al., 2014; Thakuri et al., 2014; Brahmbhatt et al., 2015; Shukla and Qadir, 2016; Mir et al., 2017). Moreover, we have also accounted for uncertainties using prevalent methods [area and length change uncertainty by Hall et al., (2003) and mapping uncertainty using buffer method by Granshaw and Fountain, (2006)] associated with glacier changes (area and length) using Landsat MSS data and also incorporated the same in the original manuscript (Table 2).

In addition to this, we have now taken 2 glaciers, GL-157 (small, 5.5 km$^2$) and Kangriz glacier (largest, 53 km$^2$) and digitized their boundaries using both the Corona and Landsat MSS imageries. On comparing the glacier boundaries using the two datasets, we noticed that higher uncertainty is associated with the GL-157 (22%) as compared to the Kangriz glacier (0.1%).Considering this, we could say that, though larger in magnitude the uncertainty estimates using Landsat would not affect GHR glaciers much (comparatively larger in size) as compared to the LR (smaller in size) glaciers.

As suggested, the Scene IDs have now been incorporated with the Table1.

Table 1 (revised manuscript): Detailed specifications of the satellite data utilized in the present study. GB= glacier boundaries, DC=debris cover

| S. no | Satellite sensors(Date of acquisition) | Remarks on quality | Scene Id | RMSE error | Registration accuracy (m) | Purpose |
|---|---|---|---|---|---|---|
| 1. | Corona KH-4B (28 Sep 1971) | Cloud free | DS1115-2282DA056/ DS1115-2282DA055/ DS1115-2282DA054 | 0.1 | 0.3 | Delineation of GB |
| 2. | LandsatMSS (19 Aug 1977/ 1 Aug 1977) | Cloud free/ peak ablation (17 Aug) | LM02_L1TP_159036_197 70819_20180422_01_T2/ LM02_L1TP_159036_197 70801_20180422_01_T2 | 0.12 | 10 | Delineation of GB, SLA&DC |
| 3. | LandsatTM (27 Aug | Partially cloud covered/ peak | LT05_L1TP_148036_1994 0827_20170113_01_T1/ | 0.22 | 6 | Delineation of GB, |

| | | | | | | |
|---|---|---|---|---|---|---|
| | 1994) | ablation | LT05_L1GS_148037_19940827_20170113_01_T2 | | | SLA&DC |
| 4. | LandsatTM (26 July 1994) | Seasonal snow cover | LT05_L1TP_148036_19940726_20170113_01_T1 | 0.2 | 6 | Delineation of GB |
| 5. | LandsatETM$^+$ (4 Sep 2000) | Cloud free/ peak ablation | LE71480362000248SGS00 | Base image | | Delineation of GB, SLA& DC |
| 6. | LandsatOLI (25July 2017) | Partially cloud covered/ peak ablation | LC08_L1TP_148036_20170810_01_T1 | 0.15 | 4.5 | Delineation of GB & DC, estimation of SLA |
| 7. | Sentinel MSI (20 Sep 2017) | Cloud free | S2A_MSIL1C_20170920T053641_N0205_R005_T43SET_20170920T053854 | 0.12 | 1.2 | Delineation of GB & DC |
| 8. | LISS IV (27Aug2017) | Cloud free | 183599611 | 0.2 | 1.16 | Accuracy assessment |

**Comment 4:** Lines 236-240: The procedures used for determining the glacier boundaries are apparently manual digitization. While this is reasonable to undertake manual processing in such complicated areas, it also necessitates a study of uncertainty estimations in such manual work. Authors may also undertake repeatability tests with different analysts to determine repeatability.

**Response 4:** We have followed a *'hybrid approach'*, involving normalized difference snow index (*NDSI*) for delineation of snow-ice boundaries and *manual digitization* for mapping the debris cover (Page: 10; lines: 231-232 of the original manuscript). Similar mapping methodology has been followed by several researchers (Chand and Sharma, 2015; Mir et al., 2017; Molg et al., 2018).

As aptly pointed out by the reviewer, we also agree that manual processing of the database necessitates uncertainty estimation. However, the essence of this work lies in the mapping of the glaciers for multiple (four) time periods by a single analyst, which minimizes the errors to a great extent. While, the repeatability tests are more relevant for studies concerning global scale inventory such as Randolph glacier inventory (RGI), Global land ice measurements from space (GLIMS) and recently Chinese glacier inventory (CGI), where multiple analysts are involved. Nevertheless, we have performed the repeatability tests on the Pensilungpa glacier by delineating its boundary for the year 2017 by 4 different analysts. The test result shows variation in glacier size by all four analysts (17.003 km$^2$, 16.22 km$^2$, 16.59 km$^2$ and 14.67 km$^2$). These values have varied significantly and slightly overestimated from the size estimated using the semi-automatic approach (15.57 km$^2$). The fluctuations in glacier size have varied within the range of 5-10%, i.e., by 9, 4, 6.5 and 6%, respectively, which is acceptable for glacier mapping (Paul et al., 2013).

**Comment 5:** Lines 272 – 300: The uncertainty assessment is biased with the very limited field validation on only one glacier for a very limited time frame. One issue that needs to be addressed is the reliability of ground truth data when different types of data were used through the nearly 50 years' time period.

**Response 5:** We agree that very limited field validation has been incorporated for a limited time frame, however, ground based monitoring of the glaciers is difficult and often constrained by extreme conditions prevailing in the Himalayan glaciated terrain. This is very well discerned from the limited field studies (11 in western, 4 in central, 1 in eastern) being conducted in the Himalayan region till date (Pratap et al., 2015; Raina and Srivastava, 2008).

In this study, the aim of comparing our results with field data (initially for 2017) was basically validating the mapping method as data related errors are being already accounted for in the other methods of uncertainty estimation. However, to enhance the reliability of ground data, we have now incorporated field data of the Kangriz glacier as well for year 2018 (obtained from DGPS). On comparing the snout position of the Kangriz glacier derived from DGPS and OLI image, an accuracy of ±1.4 m is obtained. Also, the frontal retreat estimated using DGPS and OLI image is found to be 38.63 ±*47.8* and 39.98 ±*56.6* m, respectively during the period 2017-18. This result has now been incorporated in the revised manuscript (Page: 13; lines: 320-324).

**Comment 6:** Please discuss why the projective transformation was required for the satellite data sets other than Corona?

**Response 6:** We have used projective transformation for co-registration of all the images, i.e., Landsat as well as Corona (Page: 10; lines: 227-231 of the original manuscript) in order to maintain uniformity in data processing method.

Projective transformation is a novel technique of image registration which projects the 2-dimensional image on the radius and angular coordinates, respectively. Moreover, this method has been used because in contrast to the other methods of image registration, i.e., polynomial and rubber sheeting, projective transformation involves the input reference of DEM which allows the analyst to capture the dynamics of the image and enhances the quality of the two-dimensional data.

**Comment 7:** Line 328- 330: Categorization of glaciers - is there a scientific standard for categorizing the glaciers in the different categories or was more based on the author's selectivity? Check DeBeer and Sharp (2009, Journal of Glaciology). Since the data descriptions needs to be internationally consistent, may revise.

**Response 7:** It is a welcome suggestion. However, glacier size is a variable parameter which fluctuates from basin to basin and hence, cannot be standardized globally or for a particular region. Moreover, to the best of our knowledge, there is no scientific standard for categorizing the glaciers and for this study, it is entirely based on investigators selectivity. DeBeer and Sharp, (2009) have categorized small glaciers in the British Columbia as per the size distribution of the glacier in the region, i.e., <0.4 km$^2$ as very small and 0.4-5 km$^2$ as large glaciers. However, in the Himalayan region different studies have used different size class for the glaciers (Table RT2). Owing to this heterogeneity in glacier size classification, we have not followed any particular study, but, have given a separate categorization (Page:12; lines:328-330 of the original manuscript).

Table RT2: Size distribution of glaciers in different basins of the Himalaya.

| Serial no. | Basin/ Himalayan region | Glacier size class (km$^2$) | References |
|---|---|---|---|
| 1. | Chenab, Parbati & Baspa, western Himalaya | <1
 1-5
 5-10
 >10 | Kulkarni et al., (2007; 2011) |
| 2. | Bhagirathi and Saraswati, central Himalaya | <1
 1-5
 5-10
 >10 | Bhambri et al., (2011) |
| 3. | Tista basin | <5
 5-20
 >20 | Basnett et al., (2013) |
| 4. | Koshi river basin, central Himalaya | <0.2
 0.2-0.5
 0.5-1
 1-5 | Shangguan et al., (2014) |

| | | 5-10 | |
| | | 10-20 | |
| | | >20 | |
| 5. | Ravi basin | <1 | Chand and Sharma, (2015) |
| | | 1-2 | |
| | | 2-5 | |
| | | 5-11 | |
| 6. | Drass valley, Ladakh | <1 | Koul et al., (2016) |
| | | 1-3 | |
| | | >10 | |
| 7. | Chenab basin, western Himalaya | <5 | Brahmbhatt et al., (2017) |
| | | 5-10 | |
| | | 10-20 | |
| | | >20 | |
| 8. | Central Himalaya | <5 | Garg et al., (2017) |
| | | 5-10 | |
| | | >10 | |
| 9. | Baspa basin | <0.5 | Mir et al., (2017) |
| | | 0.5-1 | |
| | | 1-5 | |
| | | 5-9 | |
| | | >9 | |
| 10. | Lidder valley, Kashmir | <1 | Murtaza and Romshoo, (2015) |
| | | 1-5 | |
| | | 5-15 | |
| 11. | Central and eastern Ladakh Himalaya | <0.25 | Schmidt and Nusser, (2017) |
| | | 0.25-0.5 | |
| | | 0.5-0.75 | |
| | | 0.75-1 | |
| | | 1-2 | |
| | | >2 | |
| 12. | Jankar Chhu watershed, Lahaul Himalaya | <0.5 | Das and Sharma, (2018) |
| | | 0.5-1 | |
| | | 1-5 | |
| | | 5-10 | |
| | | >10 | |
| 13. | Karakoram, Pamir | 0.02-0.5 | Molg et al., (2018) |
| | | 0.5-1 | |
| | | 1-5 | |
| | | 5-10 | |
| | | 10-20 | |
| | | 20-50 | |
| | | 50-100 | |
| | | >100 | |
| 14. | Miyar basin, western Himalaya | <5 | Patel et al., (2018) |
| | | >5 | |

**Comment 8:** Statistical significance could be included to explain the effect of spatial characteristics (size, aspect, debris cover) or any difference spatial control over LR and GHR.

**Response 8:** Thanks for the suggestion. We understand the reviewer's point that GHR and LR comprises of different glaciers having distinct morphology. However, in our analysis, we have taken into account the change in glacier parameters in terms of percentage, which is normalized. Hence, the data is not susceptible to any biases. Moreover, we have followed a sequential method of data analysis: in which all the glaciers are first investigated for parametric changes and we observe regional heterogeneity in glacier response. Thereafter, we went for understanding the possible controls on the reported changes, in which we noted that the glacier response is primarily influenced by climate variability (statistical significance taken into account). The study also confirms the possible controls of non-climatic factors (in terms of percentage) on heterogeneous glacier response.

However, we have now incorporated the statistical significance to explain the effect of spatial characteristics (size, slope, debris cover and elevation) over LR and GHR [Supplementary material (Text S1) of the revised manuscript]. For this, the non-climatic factors were subsequently correlated with the change in glacier dimensional parameters, i.e., area change and retreat using some statistical tests (Figure R4a,b; Table RT3). In the statistical analysis, the variables were initially tested for normality and visual inspection of the histogram. The test showed normal distribution for nearly all the variables and the correlations were found to be significant at $\alpha < 0.05$ (except for mean elevation). These correlations also showed the presence of few outliers (not removed in this study), which indicate the possible role of any other factor due to which these glaciers have deviated from the general trend of area loss and retreat (Figure R4a;b).

**Table RT3:** Correlation (r) and Pearson's correlation (p) coefficient computed between non-climatic factors (size, slope, debris cover and elevation) and glacier changes (% deglaciation & retreat rate). These relationships were found to be significant at $\alpha < 0.05$ (Except for mean elevation: Italicized).

| Parameters | % deglaciation | | Retreat rate (ma$^{-1}$) | |
|---|---|---|---|---|
| | GHR | LR | GHR | LR |
| **Size** | r= -0.385 | r= -0.5 | r= 0.475 | r= 0.524 |
| | p<0.00001 | p<0.00001 | p<0.0001 | p<0.00001 |

| Slope | r= 0.27 | r= 0.339 | r= -0.18 | r= -0.389 |
|---|---|---|---|---|
| | p<0.0022 | p<0.0002 | p<0.043 | p<0.00002 |
| *Mean elevation* | *r= -0.048* | *r= 0.002* | *r= 0.091* | *r= -0.152* |
| | *p= 0.593* | *p= 0.98* | *p= 0.31* | *p= 0.106* |
| Debris cover | r= -0.334 | r= -0.249 | r= 0.337 | r= 0.245 |
| | p<0.00013 | p<0.0075 | p<0.0001 | p<0.0088 |

[Figure]

**Figure R4a.** Scatter plots displaying the relation between topographic factors with percent deglaciation during the period 1971-2017. All the relationships were found to be significant at confidence level, i.e., α<0.05 (Except mean elevation).

[Figure]

**Figure R4b**. Scatter plots displaying the relation between topographic factors with retreat rate during the period 1971-2017. All the relationships were found to be significant at confidence level, i.e., α<0.05 (Except mean elevation).

[Figure]

Figure 3: Annual and seasonal variability in the climate data for the period 1901-2017. (a), (b) and (c) 5 year moving average of the mean annual precipitation (mm) and temperature (°C) recorded for 5 grids covering the glaciers in the entire SSB, GHR and LR (sub-regions), respectively during the period 1901-2017. The light and dark grey colored dashed lines depict the respective trend lines for precipitation and temperature conditions during the period 1901-2017. (d), (e) and (f) Monthly mean precipitation and temperature data for the entire SSB, GHR and LR (sub-regions), respectively for the time period 1901-2017.

2. Table 1 on Page 9 has been updated by incorporating the scene Id in it.

| S. no | Satellite sensors(Date of acquisition) | Remarks on quality | Scene Id | RMSE error | Registration accuracy (m) | Purpose |
|---|---|---|---|---|---|---|
| 1. | Corona KH-4B (28 Sep 1971) | Cloud free | DS1115-2282DA056/ DS1115-2282DA055/ DS1115-2282DA054 | 0.1 | 0.3 | Delineation of GB |
| 2. | LandsatMSS (19 Aug 1977/ 1 Aug 1977) | Cloud free/ peak ablation (17 Aug) | LM02_L1TP_159036_197 70819_20180422_01_T2/ LM02_L1TP_159036_197 70801_20180422_01_T2 | 0.12 | 10 | Delineation of GB, SLA&DC |
| 3. | LandsatTM | Partially cloud | LT05_L1TP_148036_1994 | 0.22 | 6 | Delineation |

| # | Sensor | Condition | Scene ID | | | Purpose |
|---|---|---|---|---|---|---|
| | (27 Aug 1994) | covered/ peak ablation | 0827_20170113_01_T1/ LT05_L1GS_148037_199 40827_20170113_01_T2 | | | of GB, SLA&DC |
| 4. | LandsatTM (26 July 1994) | Seasonal snow cover | LT05_L1TP_148036_1994 0726_20170113_01_T1 | 0.2 | 6 | Delineation of GB |
| 5. | LandsatETM⁺ (4 Sep 2000) | Cloud free/ peak ablation | LE71480362000248SGS00 | Base image | | Delineation of GB, SLA& DC |
| 6. | LandsatOLI (25July 2017) | Partially cloud covered/ peak ablation | LC08_L1TP_148036_2017 0810_01_T1 | 0.15 | 4.5 | Delineation of GB & DC, estimation of SLA |
| 7. | Sentinel MSI (20 Sep 2017) | Cloud free | S2A_MSIL1C_20170920T 053641_N0205_R005_T43 SET_20170920T053854 | 0.12 | 1.2 | Delineation of GB & DC |
| 8. | LISS IV (27Aug2017) | Cloud free | 183599611 | 0.2 | 1.16 | Accuracy assessment |

3. Line stating "The glacierets/ tributary glaciers contributing to the main trunk are considered as single glacier entity" has been added on Page 10; lines: 239-240.

4. On page 11; lines: 268-270 have been updated for calculation of "change in climate variables" instead of percentage change.

5. On pages: 11-13; lines: 276-308 have been incorporated Further, in order to check data consistency, we have taken up instrument data from nearest stations of Kargil and Leh (due to the unavailability of meteorological stations in the Suru sub-basin) and compared with the CRU-TS derived data for the entire Suru sub-basin during 1901-2002 period (Figure 4).

[revised manuscript text omitted]

11. We have now reported the temperature and precipitation changes in absolute terms rather than percentage on Pages: 19,20 & 25 in track change mode.

12. Greater Himalayan Range (GHR) and Ladakh Range (LR) comprises of different glaciers having distinct morphology. Therefore, statistical significance becomes necessary to explain the effect of spatial characteristics (size, slope, debris cover and elevation) over LR and GHR. For this, the non-climatic factors were subsequently correlated with the change in glacier dimensional parameters, i.e., area change and retreat using some statistical tests (Figure R4a,b; Table RT2). In the statistical analysis, the variables were initially tested for normality and visual inspection of the histogram. The test showed normal distribution for nearly all the variables and the correlations were found to be significant at $\alpha < 0.05$ (except for mean elevation). These correlations also showed the presence of few outliers (not removed in this study), which indicate the possible role of any other factor due to which these glaciers have deviated from the general trend of area loss and retreat (Figure R4a;b).

**Table RT2:** Correlation (r) and Pearson's correlation (p) coefficient computed between non-climatic factors (size, slope, debris cover and elevation) and glacier changes (% deglaciation & retreat rate). These relationships were found to be significant at $\alpha < 0.05$ (Except for mean elevation: Italicized).

| Parameters | % deglaciation | | Retreat rate (ma⁻¹) | |
|---|---|---|---|---|
| | **GHR** | **LR** | **GHR** | **LR** |
| **Size** | r= -0.385 | r= -0.5 | r= 0.475 | r= 0.524 |
| | p<0.00001 | p<0.00001 | p<0.0001 | p<0.00001 |
| **Slope** | r= 0.27 | r= 0.339 | r= -0.18 | r= -0.389 |
| | p<0.0022 | p<0.0002 | p<0.043 | p<0.00002 |
| *Mean elevation* | *r= -0.048* | *r= 0.002* | *r= 0.091* | *r= -0.152* |
| | *p= 0.593* | *p= 0.98* | *p= 0.31* | *p= 0.106* |
| **Debris cover** | r= -0.334 | r= -0.249 | r= 0.337 | r= 0.245 |
| | p<0.00013 | p<0.0075 | p<0.0001 | p<0.0088 |

[Figure]

**Figure R4a.** Scatter plots displaying the relation between topographic factors with percent deglaciation during the period 1971-2017. All the relationships were found to be significant at confidence level, i.e., α<0.05 (Except mean elevation).

[Figure]

**Figure R4b**. Scatter plots displaying the relation between topographic factors with retreat rate during the period 1971-2017. All the relationships were found to be significant at confidence level, i.e., α<0.05 (Except mean elevation).

14. Reference list has been updated.

[revised manuscript text omitted]

---

## Author Response (AR2)

Dear Editors,

Please find enclosed the marked-up revised manuscript (**Ref. Manuscript ID:essd-2019-122**)

entitled **"Temporal inventory of glaciers in the Suru sub-basin, western Himalaya:**

**Impacts of the regional climate variability".** On behalf of all the authors, I would like to convey my sincere thanks to the topical editor, the editorial and administrative team of the

Earth System Science Data for timely processing of the manuscript and suggesting constructive comments for improving the original manuscript substantially. In line with the suggested corrections (mainly grammatical errors) we have edited the text and revised the manuscript accordingly.

Thanks for your consideration.

Yours Sincerely,

Aparna Shukla.

[revised manuscript text omitted]